# Local Spatiotemporal Representation Learning for Longitudinally-consistent Neuroimage Analysis

**Mengwei Ren**
New York University
`mengwei.ren@nyu.edu`

**Neel Dey**
New York University
`neel.dey@nyu.edu`

**Martin A. Styner**
UNC-Chapel Hill
`styner@cs.unc.edu`

**Kelly N. Botteron**
WUSTL School of Medicine
`botteronk@wustl.edu`

**Guido Gerig**
New York University
`gerig@nyu.edu`

## Abstract

Recent self-supervised advances in medical computer vision exploit the global and local anatomical self-similarity for pretraining prior to downstream tasks such as segmentation. However, current methods assume i.i.d. image acquisition, which is invalid in clinical study designs where follow-up *longitudinal* scans track subject-specific temporal changes. Further, existing self-supervised methods for medically-relevant image-to-image architectures exploit only spatial or temporal self-similarity and do so via a loss applied only at a single image-scale, with naive *multi-scale spatiotemporal* extensions collapsing to degenerate solutions. To these ends, this paper makes two contributions: (1) It presents a *local* and *multi-scale* spatiotemporal representation learning method for image-to-image architectures trained on longitudinal images. It exploits the spatiotemporal self-similarity of learned multi-scale intra-subject image features for pretraining and develops several feature-wise regularizations that avoid degenerate representations; (2) During finetuning, it proposes a surprisingly simple self-supervised segmentation consistency regularization to exploit intra-subject correlation. Benchmarked across various segmentation tasks, the proposed framework outperforms both well-tuned randomly-initialized baselines and current self-supervised techniques designed for both i.i.d. and longitudinal datasets. These improvements are demonstrated across both longitudinal neurodegenerative adult MRI and developing infant brain MRI and yield both higher performance and longitudinal consistency.

## 1 Introduction

Tracking *subject-specific* anatomical trends over time is crucial to both clinical diagnostics and large-scale biomedical science. Such *longitudinal imaging* is especially relevant to analyzing neurological patterns of growth and degeneration via brain imaging in pediatric and elderly populations, respectively. As tracking individual structural changes requires precise and longitudinally-consistent segmentation methods with scarce annotated training volumes, we identify two major bottlenecks in existing self-supervised biomedical image analysis methods which we use to motivate our work.

**Learning with few annotations.** While modern imaging studies may scan hundreds to thousands of individuals, manually outlining volumetric structures of interest across multiple individuals for supervised segmentation training is prohibitively expensive. Therefore, current work focuses on leveraging large sets of unlabeled images to *pretrain* image-to-image architectures (e.g., the U-Net [48]), which can then be efficiently finetuned in the one or few-shot setting. These *self-supervised* methods may handcraft pre-text training objectives [10, 19, 45, 65] or may attempt to pretrain the base

36th Conference on Neural Information Processing Systems (NeurIPS 2022).

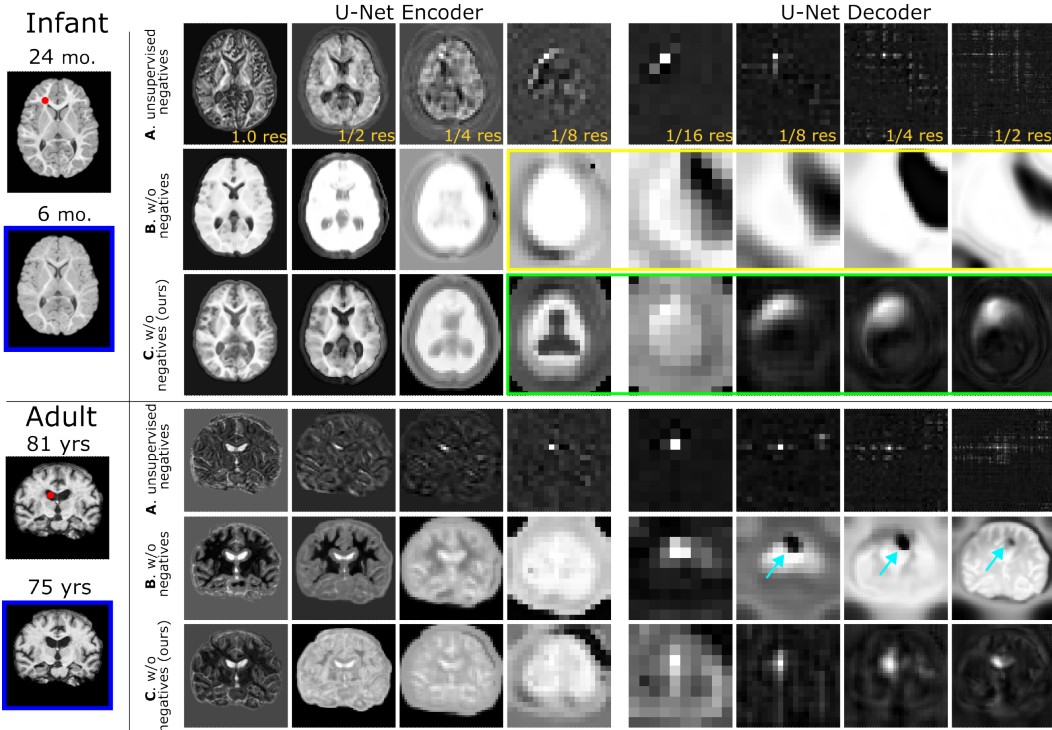

Figure 1: On pretraining an image-to-image network with *per-layer* spatiotemporal self-supervision, we visualize the **intra-subject multi-scale feature similarity** between a **query** channel-wise feature and all spatial positions within the **key** feature at a different age. **A:** Contrastive pretraining with unsupervised negatives [44] yields only positionally-dependent representations. **B:** Pretraining w/o negatives [11] by using corresponding intra-subject patch locations as positives leads to semantically-implausible representations with low-diversity (e.g., see yellow box) and artifacts (see arrows) in deeper layers. **C:** Our method attains both positionally and anatomically-relevant representations via proper regularization (e.g., see green box). Additional structures are visualized in Suppl. Figure 5.

network to be equivariant to transformationsin order to preserve semantic meaning as in contrastive learning in medical [9, 63] and natural image vision [2, 25, 38, 60, 67, 68]. However, these methods typically leverage label supervision in sampling for their losses and impose a self-supervised loss only at a single feature scale (typically the encoder bottleneck or network output). Naive application of local unsupervised multi-scale contrastive losses [16, 44] lead to degenerate representations (Fig. 1, row A) when applied to both the encoder and decoder of image-to-image architectures.

**Violating i.i.d. assumptions.** Further, most existing self-supervised frameworks assume i.i.d. data. Unfortunately, this assumption does not transfer to longitudinal studies where intra-subject temporal images are highly correlated. Emerging longitudinal representation learning methods focus on imposing temporal-consistency into the encoder bottleneck [15, 43, 66], such that the encoder learns representations that are aware of the order of acquisition [15] and the overall trajectory [43]. These methods address image-level tasks such as disease classification or progression and age prediction. However, their extension to pixel-level applications with image-to-image architectures remains unclear.

**Methods.** Motivated by the above limitations, in this work, we claim that the spatiotemporal dependency of imaging data should be explicitly incorporated in self-supervised frameworks. We do so by exploiting the spatial and temporal self-similarity of local multiscale deep features in both encoder and decoder and further learn diverse intermediate representations by developing regularizations for self-supervised similarity objectives. Lastly, when finetuning with limited annotated data, we encourage predictions on unlabeled subject-wise images to be spatiotemporally consistent.

**Contributions.** This work makes the following contributions: (1) It presents a longitudinally-consistent spatiotemporal representation learning framework to learn from image time-series; (2) To impose multi-scale local self-supervision while avoiding degenerate solutions, it develops regulariza-

tion terms on the variance, covariance, orthogonality of local features within the decoder; (3) To further self-supervise the fine-tuning stage, the proposed method encourages segmentations from adjacent timepoints on unlabeled data to be consistent; (4) Across three large-scale longitudinal one, few, and full-shot segmentation tasks on both elderly and pediatric populations, the developed framework yields improved segmentation performance and higher longitudinal segmentation consistency. Our code is available at https://github.com/mengweiren/longitudinal-representation-learning.

## 2 Related work

**Self-supervision.** Self-supervised learning (SSL) methods aim to learn hierarchical representations from unannotated data, which can then be transferred to tasks operating in low-annotation regimes. Early work focused on pretext tasks where a handcrafted loss is used to pretrain networks via orientation prediction [19], context restoration [10, 45], channel prediction [65], among others. Given their heuristic nature and suboptimal generalization, recent work instead focuses on data-driven SSL losses.

**Contrastive learning.** Contrastive learning [11, 22, 23, 24, 49, 55] (CL) typically transforms an input image and asks the embeddings of the input image and its transformation (the *positive* pair) to be close to one another and far apart from embeddings of other images (*negatives*) via a noise contrastive estimation [21, 42] (NCE) loss. While performant on image-level recognition tasks [12, 32], CL requires non-trivial modification to extend to pixel-level segmentation tasks, as described below.

**Negative-free representation learning.** In several applications, *true* negative samples may be difficult to construct [26]. For example, when learning on intra-domain internal image patches [16, 44], non-local spatial positions may be semantically similar, but NCE objectives push their embeddings apart, leading to false negative pairs introducing label noise in the training objective. This drawback may be mitigated via data-driven SSL methods which only use positive samples and avoid low-diversity (or *collapsed* [30]) embeddings solutions via predictor networks and custom backpropogation [13, 20] and careful regularization [6, 62]. However, as above, these methods operate on *global* image embeddings and require modification for pixel-level tasks.

**Spatial self-supervision.** Towards downstream segmentation, recent work [2, 25, 35, 38, 60, 67, 68] encourages local single-scale features either within an image or across images to cluster semantically by constructing positive pairs using ground-truth labels. To incorporate *local* and *multi-scale* spatial considerations into unpaired image translation and registration [16, 44] imposed contrastive losses on randomly-sampled layer-wise encoder features by considering corresponding spatial indices as positives and all other locations as negatives. Our work builds on this by instead only considering temporal positives (described below) in the layerwise losses alongside custom regularization which avoids low-diversity decoder embeddings observed with naive application in Fig. 1.

**Temporal self-supervision.** SSL methods developed for video achieve high performance by exploiting temporally consistent transformation [5] and temporal pretext task [14, 29, 59]. However, longitudinal biomedical image time-series have sparser sampling (typically 2–5 timepoints/subject) and have greater spatial extents (volumes instead of images), which leads to distinct modeling considerations.

Emerging biomedical methods [15, 43, 66] enforce smooth trajectories for subject-wise images in the encoder latent space and deploy their methods on image-level downstream tasks such as disease classification and age regression. However, these methods focus on learning a global embedding, without a clear extension to pixel-level tasks such as segmentation.

**Biomedical image segmentation.** Major challenges specific to biomedical segmentation include: (1) large 3D volumes; (2) limited sample sizes and annotations; and (3) non-i.i.d. longitudinal acquisitions tracking temporal anatomical changes. To these ends, conventional approaches use a combination of intensity-based probabilistic models and registration-driven atlas-based models [1, 27, 39, 58]. In particular, longitudinal image analysis typically makes use of one or few longitudinal atlases [28, 33, 47, 51, 52], which motivates the one and few-shot segmentation settings benchmarked in this paper, respectively.

More recently, deep segmentation networks achieve strong performance [8, 7, 40, 41, 48] given enough training volumes. In the low-annotation setting, weakly supervised methods develop custom loss functions [31, 37], but may have drawbracks analogous to the handcrafted SSL losses described above. Fortunately, recent data-driven self and semi-supervised methods are well-suited to pixel-level prediction. For example, to pretrain an encoder for segmentation, [9, 63] develop

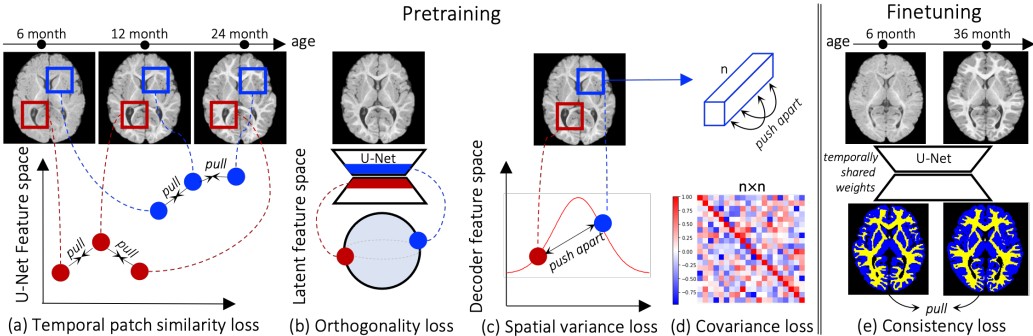

Figure 2: **Overview of proposed self-supervision.** Given nonlinearly-registered temporal images of a subject, **(a)** we assume that corresponding spatial locations in various network layers should have similar representations. As U-Net skip connections can cause degenerate decoder embeddings (see App E), we **(b)** encourage the decoder bottleneck to be orthogonal to encoder bottleneck and regularize the concatenated decoder features to have **(c)** high spatial variance and be **(d)** uncorrelated channel-wise. During fine-tuning, we **(e)** encourage temporal intra-subject network output consistency.

application-specific positive and negative sampling strategies for contrastive training, where 2D slices from similar locations in registered 3D volumes across subjects constitute positive pairs. While these methods have been successful in their applications, they are inherently slice-based methods and are outperformed by well-tuned randomly-initialized 3D baselines on our datasets (Tab. 1). Further, these self-supervised biomedical segmentation methods do not explicitly account for non i.i.d acquisitions. Lastly, to our knowledge, existing longitudinal deep learning work developed for biomedical segmentation is currently very specific to its target application [18, 36, 56] (for example, in tasks such as MS lesion change detection [56]) or require supervised pre-training on annotated cross-sectional datasets [61], whereas we develop a generic self-supervised spatiotemporal representation learning framework for non-i.i.d. longitudinal data which can be applied to any downstream task in principle.

## 3 Methodology

Fig. 2 illustrates the proposed framework. The base U-Net architecture is pretrained end-to-end in a self-supervised manner with intra-subject spatiotemporal losses. On convergence, the pretrained network parameters serve as an initialization for training downstream local spatiotemporal pixel-level tasks (e.g., registration or segmentation). This work will focus on downstream segmentation in the one, few, and full-shot regimes. The main similarity loss is applied on multiscale local patches from different timepoints of the same subject, which attracts features in corresponding locations in separate timepoints together. The high-dimensional U-Net features corresponding to the boxes at the same locations in Fig. 2a should maintain high similarity despite varying appearance. The feature regularizers (Fig. 2b,c,d) avoid degenerate U-Net decoder embeddings in the patch similarity training and the output regularizer (Fig. 2e) encourages finetuning consistency on unannotated data.

**Setup.** The unlabeled dataset is a collection of $N$ subjects, where each subject has at least two longitudinal image acquisitions available during pretraining. During every iteration, a pair of images $(\mathcal{X}_j^i, \mathcal{X}_{j+1}^i)$ are randomly sampled from subject $i \in \{1, 2, \dots, N\}$ at distinct timepoints $j$ and $j + 1$, where $j \in \{1, 2, \dots, T_i - 1\}$. $T_i$ indicates the number of registered images from subject $i$, and $\mathcal{X} \in \mathcal{R}^{W \times H \times D \times C}$ is a 3D volume of spatial dimension $W \times H \times D$ and $C$ channels. These channels are typically multi-modality acquisitions (e.g., T1w and T2w MRI from the same subject).

**Spatiotemporal patchwise similarity loss.** We aim to associate the local embeddings of $\mathcal{X}_j^i$ and $\mathcal{X}_{j+1}^i$ such that the representations are longitudinally-aware. To this end, a weight-sharing 3D U-Net $G$ takes both images as input and produce a set of multi-scale CNN features $\{v\}_L$, where each element $v_{lij} = G^l(x_j^i) \in \mathcal{R}^{W_l \times H_l \times D_l \times C_l}$ indicates the output of the $l$th layer of interest. $M$ feature vectors are then randomly sampled from the 3D spatial indices of the feature map, where each feature vector $v_{lij}^m$ represents a local patch of the input image. These patch-wise activations from multiple layers of $G$ form hierarchical representations of local regions of the input image.

To maximize the similarity of corresponding local features (e.g., blue boxes in Fig. 2a) without negative samples, we use a projector MLP $f$ and a predictor head $p$ and extend [13] to patchwise operation such that matching spatial indices have high agreement. The patchwise similarity loss between two representations is defined as follows: $\mathcal{L}(v_{lij}^m, v_{lik}^m) = \frac{1}{2}\mathcal{D}(p_1, z_2) + \frac{1}{2}\mathcal{D}(p_2, z_1)$, where $\mathcal{D}(p, z) = -\frac{p}{||p||_2} \cdot \frac{z}{||z||_2}$, $z_1, z_2 = f(v_{lij}^m), f(v_{lik}^m)$ and $p_1, p_2 = p(z1), p(z2)$. The total loss is an average of all sampled patches, across multilayer features:

$$\mathcal{L}_{sim} = \frac{1}{L} \sum_{l \in \{1,2,...L\}} \frac{1}{M} \sum_{m \in \{1,2,...M\}} \mathcal{L}(v_{lij}^m, v_{lik}^m). \tag{1}$$

**Architectural challenges in multi-scale representation learning**. $\mathcal{L}_{sim}$ applied to the hidden layers of a U-Net is found to maximize patchwise similarity of the encoder features (as expected) but lead to low-diversity and semantically-incoherent representations in the decoder layers as observed from the similarity maps in Fig. 1B and empirical observations in App. E. We *speculate* that this is partially attributable to U-Net skip connections, which lead to a degenerate solution where the encoder learns representations which are good enough to minimize the decoder losses and the decoder layers do not have to learn useful representations which transfer. To this end, we develop several regularization strategies such that the decoder layers obtain diverse and semantically-coherent embeddings.

**Orthogonality.** During model prototyping, we empirically observe that low-diversity embeddings first originate in the U-Net bottleneck (Fig. 1B cols 4,5 and App. Fig. 14 rows 2,3) which are then upsampled hierarchically through the decoder. We therefore encourage decoupled bottleneck features between the encoder and decoder. Revisiting the U-Net skip-connection, the encoder features from the $l$-th layer $v_e \in \mathcal{R}^{W_l \times H_l \times D_l \times C_l}$ are concatenated with the upsampled decoder features $v_d \in \mathcal{R}^{W_l \times H_l \times D_l \times C_d}$, followed by a convolution to attain the same feature dimension as $v_e$. This yields $v_{\hat{d}} = \texttt{Conv}(\texttt{Concat}(v_e, v_d)) \in \mathcal{R}^{W_l \times H_l \times D_l \times C_l}$ and by regularizing for orthogonality between $v_e$ and $v_{\hat{d}}$ in the projector embedding space, we implicitly encourage $v_d$ to learn better representations instead of converging to degenerate solutions. The orthogonality loss is defined as

$$\mathcal{L}_O(z_e, z_{\hat{d}}) = \frac{1}{M} \sum_{m \in \{1,...M\}} \frac{z_e^m}{||z_e^m||_2} \cdot \frac{z_{\hat{d}}^m}{||v_{\hat{d}}^m||_2}, \tag{2}$$

where $z_e^m$ and $z_{\hat{d}}^m$ are projected representations via $f$ at sampling location $m$, from the matched encoder/decoder layers, respectively.

**Variance and covariance.** We further encourage spatial variation and channel-wise decorrelation of local decoder features to avoid degenerate representations. We extend the variance and covariance terms of [6] towards patchwise multi-layer operation and apply them to the decoder layers. A standard deviation loss encourages spatial feature variation above a threshold of $\eta = 1$ and is defined as

$$\mathcal{L}_S(z) = \frac{1}{k} \sum_{l \in \{L-k,...L\}} (\max(0, \eta - S(z_l)), \tag{3}$$

where $S(z_l) = \sqrt{Var(z_l) + \epsilon}$ is the standard deviation of $M$ randomly-sampled projected features from the $l$-th layer and $\epsilon = 10^{-4}$ is added for numerical stability. A covariance regularization decorrelates channelwise activations in the decoder to prevent low-diversity embeddings by minimizing

$$\mathcal{L}_C(z) = \frac{1}{k} \sum_{l \in \{L-k,...L\}} \frac{1}{n} \sum_{u \neq v} [C(z)]_{u,v}^2, \tag{4}$$

where $C(z) = \frac{1}{n-1} \sum_i^n (z_i - \overline{z})(z_i - \overline{z})^T$ is the covariance matrix of $n$-D representation $z$ and $(u, v)$ indicates its off-diagonal indices.

**Reconstruction.** To further pretrain the decoder, we investigate reconstruction losses commonly used in unsupervised learning. However, as high-resolution information is passed through skip connections, a reconstruction loss for a U-Net is near-trivially minimized and does not address degenerate solutions. Therefore, inspired by denoising autoencoders [57], we encourage network equivariance and invariance to geometric ($\mathcal{A}_g$) and intensity-based transformations ($\mathcal{A}_i$), respectively, of the input image $x$, by modifying the reconstruction objective to a denoising loss as,

$$\mathcal{L}_{rec} = ||G(\mathcal{A}_i(\mathcal{A}_g(x))) - \mathcal{A}_g(x)||_2^2. \tag{5}$$

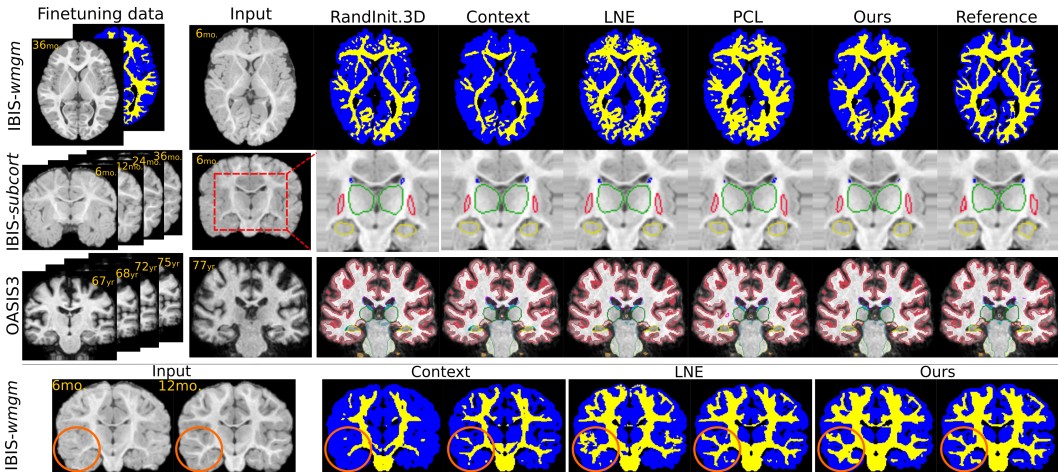

Figure 3: **One-shot segmentation**. **Top 3 rows:** Once pretrained on all unlabeled data, all benchmarked methods are finetuned on either a single annotated image (IBIS-*wmgm*) or a single annotated subject (IBIS-*subcort* and OASIS3). When deployed on other subjects at different ages, our method yields improved segmentation performance. **Bottom row:** When finetuned only on a single 36 month-old image, our method generalizes to unseen timepoints by leveraging temporal consistency.

**Finetuning with longitudinal consistency regularization.** In longitudinal segmentation, if the finetuning data do not cover the overall age range, the network may perform well on the finetuned timepoints, but may perform poorly on unseen ages even when pretrained on unannotated data across all ages. We therefore develop a self-supervised longitudinal consistency regularization term applied at the network output during finetuning to increase the intra-subject agreement. Given registered and unannotated image volumes, we formulate a segmentation prediction consistency loss as,

$$\mathcal{L}_{cs} = 1 - \texttt{Dice}(G(x_j^i), G(x_{j+1}^i)), \tag{6}$$

which is minimized alongside the supervised segmentation term below during finetuning.

**Total objective.** During pretraining, our overall objective function is a weighted sum of the patch similarity loss, reconstruction loss, and three regularizations, and is defined as,

$$\mathcal{L}_{PT} = \lambda\mathcal{L}_{sim} + \alpha\mathcal{L}_{rec} + \mu\mathcal{L}_S(z) + \gamma\mathcal{L}_C(z) + \beta\mathcal{L}_O(z). \tag{7}$$

During finetuning, we use a combined dice and cross entropy loss on supervised training pairs as,

$$\mathcal{L}_{sup} = (1 - \texttt{Dice}(G(x), y)) + \texttt{CE}(G(x), y), \tag{8}$$

where $y$ is the groundtruth segmentation label. Lastly, we additionally use the segmentation consistency loss $\mathcal{L}_{cs}$ for non i.i.d inputs, which forms our final finetuning loss $\mathcal{L}_{FT} = \mathcal{L}_{sup} + \mathcal{L}_{cs}$.

## 4 Experiments

**Data and segmentation tasks.** We conduct experiments on two de-identified longitudinal neuroimaging datasets, and specifically design three tasks to benchmark different extents of biomedical domain gaps between the finetuning and testing data. The main body of this work focuses on one-shot segmentation using one annotated subject (a common medical image analysis setting). Benchmarks on few-shot and fully-supervised segmentation tasks are provided in Appendix A. For both datasets, we perform a train/validation/test split on a subject-wise basis with 70%, 10% and 20% of the participants. The validation set is used for model and hyperparameter selection and results are reported on a held-out test set. ANTs [3, 4] is used to perform inter-subject affine alignment for all experiments, followed by intra-subject deformable registration to obtain accurate spatiotemporal correspondence for $\mathcal{L}_{sim}$ and $\mathcal{L}_{cs}$ calculation. All images are skull-stripped, bias-field corrected, and intensity-normalized. Further data preprocessing and splitting details are described in Appendix C.

OASIS3 [34] is a publicly-available dataset consisting of 1639 brain MRI scans of 992 longitudinally imaged subjects. Each subject has 1–5 temporal acquisitions over a $\sim$ 5-year long observation

window, resulting in an aging cohort over the span 42 to 95 years which includes cognitively normal and mildly impaired individuals alongside subjects with Alzheimer's Disease. On OASIS3, we tackle whole-brain segmentation using the FreeSurfer label convention [17]. *Cross-sectional* FreeSurfer anatomical segmentation was done as part of the data release. Observing strong temporal inconsistency (see App. C.2), we further perform *longitudinal* FreeSurfer [47] to improve the temporal consistency of the reference segmentation. We exclude labels that have less than 100 voxels in all subjects, which results in 33 labels for segmentation training and evaluation. Finetuning for one-shot segmentation is performed on a single FreeSurfer-annotated subject with four timepoints.

IBIS is an infant brain imaging study, which longitudinally acquires 1272 structural T1w/T2w MRI from 552 infants across both controls and infants at a high-risk for Autism Spectrum Disorder (ASD) over a span of 3 to 36 months of age. We tackle two distinct tasks: subcortical segmentation (*IBIS-subcort*) and white/gray matter tissue segmentation (*IBIS-wmgm*). For *IBIS-subcort*, a multi-atlas method [54] cross-sectionally segments sub-cortical grey matter (relevant to ASD [49]) into 13 structures of interest, which are then followed by manual corrections. For one-shot benchmarking, finetuning is performed only on a single longitudinally-labeled subject, similar to OASIS3 above. Further, we use the *IBIS-wmgm* setting to simulate a real-world use-case detailed in App. D.2. Briefly, brain MRI segmentation into grey/white matter is straightforward at 24-36 months of age due to the presence of anatomical edges in the images. However, ∼6-month-old grey/white matter brain segmentation remains elusive without manually labeled datasets for supervision due to white matter myelination leading to isointense appearance at that age [53] (e.g., Fig. 2a and App. D.2). We therefore investigate finetuning all benchmarked methods on a single 36 month old image which can be reliably segmented with [46] and then evaluate segmentation deployment with a strong domain shift on 6 month old isointense images and labels (whose ground truth labels are generated by a fully supervised external model [64]).

**Baselines and Evaluation Strategies.** We analyze segmentation performance and longitudinal consistency against well-tuned randomly initialized 2D/3D U-Nets (`RandInitUnet.2D/3D`) and various high-performing self-supervised pretraining methods. These include the pretext-task based `Context Restoration` [10], longitudinal representation learning based `LNE` [43], along with the contrastive learning based `GLCL` [9] and `PCL` [63] methods which operate on image slices. We also repurpose `PatchNCE` [44] for segmentation to evaluate its generic representation learning capabilities. All methods are pretrained and finetuned with both geometric and intensity-based augmentation, and share the same network architecture.

We quantify network performance via commonly used scores such as the Dice coefficient, IoU, and the 95-th percentile of the Hausdorff distance. More importantly, we also quantify the longitudinal agreement between intra-subject non-linearly registered temporal segmentations via scores such as the spatiotemporal consistency of segmentation [36] $STCS = \frac{2|S_1 \cap S_2|}{|S_1| + |S_2|}$ where $S_1$ and $S_2$ are temporal segmentation predictions from non-linearly registered input images; and the absolute symmetrized percent change [47] $ASPC = 100 \frac{|V_2 - V_1|}{0.5(V_1 + V_2)}$, where $V_1$ and $V_2$ are the volume of a structure calculated from $S_1$ and $S_2$. We also report $STCS$ and $ASPC$ on the groundtruth segmentations as a reference.

**Implementation details.** We train a 3D U-Net [48] as the base image-to-image architecture with four levels of up/down sampling and repeated Conv-BN-ReLU blocks (all architectural details are provided in App. B). The projector head consists of a 3-layer MLP with 2048 nodes per layer. Following [13], we apply batch normalization after each MLP, followed by ReLU activation, and $l_2$ normalization of the final activation. The predictor is a 3-layer bottlenecked MLP with widths of 2048-256-2048. We apply geometric (left-right flip, random affine warps) and intensity augmentations including random blur, noise, gamma contrast enhancement. Additional MRI-specific augmentations with random bias field and motion artifacts are also applied and are followed by $128^3$ random spatial cropping. We use a batch size of 3 crops and an initial learning rate of $2 \times 10^{-4}$ for both pretraining and finetuning. All networks are trained with the Adam optimizer ($\beta_1 = 0.9$ during pretraining and $\beta_1 = 0.5$ during finetuning and $\beta_2 = 0.999$ in both settings) on a single Nvidia RTX8000 GPU (45GB vRAM). The networks are pretrained for a maximum of $30,000$ steps and the best model based on validation performance is used for fine-tuning for another $35,000$ steps, alongside linear learning rate decay. All experiments are run on a fixed random seed due to limited computational budgets. Based on the ablation analysis in Tab. 2, we empirically choose $\lambda = 1, \alpha = 10, \gamma = 1e-3, \beta = 100$ for all datasets, and use $\mu = 10^{-2}$ for OASIS3, $\mu = 10^{-3}$ for IBIS. Further details on the configurations, implementation of our method and other baselines are provided in Appendix B.

Table 1: **One-shot segmentation benchmarking** of performance (median IoU & HD95) and longitudinal consistency (ASPC). Further scores alongside means and std. dev. are provided in Supplemental Table 7. **Few-shot** and **fully-supervised** results are provided in Suppl. Tabs. 3 and 4, respectively.

| Method | IBIS-subcort | | | IBIS-wmgm | | | OASIS3 | | |
|---|---|---|---|---|---|---|---|---|---|
| | IoU ↑ | HD95↓ | ASPC↓ | IoU↑ | HD95↓ | ASPC↓ | IoU ↑ | HD95↓ | ASPC↓ |
| GT | - | - | 7.819 | - | - | - | - | - | 3.947 |
| RandInitUnet.2D | 0.707 | 4.485 | 7.127 | 0.510 | 3.274 | 11.506 | 0.687 | 2.545 | 10.189 |
| RandInitUnet.3D | 0.720 | 2.892 | 4.644 | 0.560 | 3.788 | 5.515 | 0.715 | 2.206 | 2.789 |
| Context Restore [10] | 0.711 | 4.403 | 7.831 | 0.444 | 8.273 | 29.235 | 0.717 | 3.323 | 5.577 |
| LNE [43] | 0.736 | 3.033 | 5.866 | 0.563 | 3.201 | 5.352 | 0.726 | 1.988 | 8.836 |
| GLCL [9] | 0.718 | 3.203 | 5.514 | 0.550 | 4.112 | 8.472 | 0.695 | 2.264 | 4.622 |
| PCL [63] | 0.713 | 3.270 | 5.610 | 0.562 | 4.974 | 10.648 | 0.707 | 2.327 | 4.850 |
| PatchNCE [44] | 0.743 | 1.266 | 5.780 | 0.607 | 4.344 | 3.782 | 0.738 | 2.275 | 3.114 |
| Ours w/o $\mathcal{L}_{cs}$ | 0.754 | **1.145** | 5.483 | 0.614 | 3.291 | **2.462** | **0.739** | **1.940** | **2.729** |
| Ours w/ $\mathcal{L}_{cs}$ | **0.757** | 1.178 | **4.475** | **0.676** | **3.237** | 4.155 | 0.737 | 2.094 | 2.754 |

Figure 4: **One-shot segmentation benchmarking** quantifying performance with the Dice coefficient (**top**) and the spatiotemporal consistency of segmentation (**bottom**), visualizing the means and standard deviations alongside median values overlaid on the top of each subfigure (higher is better). **Few-shot** and **fully-supervised** results are provided in Suppl. Tabs. 3 and 4, respectively.

**Segmentation and longitudinal consistency results.** Fig. 3 qualitatively demonstrates improved generalization using our method on unseen longitudinal data (row 1-3), especially on data displaying rapid intra-subject temporal developments (IBIS-*wmgm,subcort*). These improvements are consistent with the quantitative results presented in Fig. 4/Tab. 1 which indicate both improved segmentation performance and longitudinal consistency. In the strong domain shift setting of *IBIS-wmgm*, we see a near ten-point increase in median dice over most baselines. With moderate shifts in *IBIS-subcort*, we see appreciable increases in performance and consistency. We note that brain segmentation on adult brain MRI data from OASIS3 is a comparatively easier task as adult neuroimages do not significantly change appearance between imaging sessions. Therefore, several baselines are able to match (but not exceed) the segmentation performance of our method on OASIS3. However, all baselines are outperformed by ours on all datasets in terms of longitudinal-consistency which is essential to non-i.i.d. statistical analysis. In particular, both our pretraining (`Ours w/o` $\mathcal{L}_{cs}$) and finetuning (`Ours w/` $\mathcal{L}_{cs}$) methods show STCS and ASPC improvements over all of the compared settings. Fig. 3 (bottom row) shows an example of temporal predictions on two unseen timepoints from IBIS-wmgm. The predictions from Context Restoration [10] match only the input image intensity and lack anatomical and longitudinal consistency, LNE [43] introduces false positive predictions in temporal lobe (within the orange circle), and the proposed method yields a more spatiotemporally and anatomically

consistent segmentation. Beyond one-shot segmentation, we also observe gains in the **few-shot** and **fully-supervised** segmentation settings in Appendix A and Suppl. Tabs. 3 and 4, respectively.

**Qualitative self-supervised spatiotemporal similarity.** In Fig. 1, we qualitatively examine the learned visual representations of the proposed method via intra-subject temporal self-similarity (**C**) and compare it to two of its variants which either use contrastive learning with unsupervised negatives (A) or negative-free representation learning (B). We calculate the per-layer multiscale feature self-similarity between the query and each key from the intra-subject feature maps at a different age (blue box). In row A, we see that assuming that all spatial indices not in correspondence constitute negative pairs leads to highly-positionally dependent representations in the decoder which carry low semantic meaning (e.g., in the adult data, the similarity to localize to the ventricles in the coronal view). By discarding all negatives in row B, we observe semantically-incoherent and low-diversity embeddings and artifacts in the decoder layers on both datasets. Finally, with careful regularization in row C, our methods discards all negative pairs and attains semantically and positionally relevant representations.

**Ablations.** As the proposed method consists of several moving parts, an ablation analysis is conducted over different model configurations, hyperparameters, and loss functions, reported in Tab. 2 consisting of average dice coefficients. The combination of all proposed components yields optimal results. Further ablations and baseline tuning results are reported in Appendix A.

Row A starts with a base setting where only four encoder layers from the U-Net are selected for $\mathcal{L}_{sim}$ computation and a small MLP width of 256 is used for the projection and prediction heads. Here, IBIS-subcort and OASIS3 results are competitive with randomly initialized U-Net, as expected given the lack of auxiliary losses, data augmentation, and regularizations. However, the IBIS-wmgm experiment already shows a 2% improvement over random initialization, indicating benefits of using patchwise similarity losses for better out-of-distribution generalization even with suboptimal setups.

Row B: With larger projector and predictor networks, we observe improvements on two out of three datasets, which is consistent with trends observed on natural images [12, 13].

Rows C–F: On adding decoder layers to $\mathcal{L}_{sim}$ and introducing $\mathcal{L}_{rec}$ alongside data augmentation (without any regularization), we typically observe inconsistent dataset-specific trends which arise from unregularized representations (e.g., Fig. 1B). We speculate that a poorly trained decoder (due to a lack of regularization) may be equivalent to random initialization in the context of pretraining for segmentation tasks. However, a combination of these components (Row F) leads to an appreciable increase in performance.

Rows G–K: When orthogonal regularization and/or covariance/variance regularization is used, we observe the best performance when they are applied together alongside augmentation and $\mathcal{L}_{Rec}$. In rows J and K, we observe that different hyperparameters are optimal for OASIS3 and IBIS (which already outperform all baseline methods in Tab. 1), which is intuitive as these are drastically different cohorts.

Row L: Finally, the overall proposed model is achieved when $L_{cs}$ is added to the finetuning objective, which yields strong improvements for IBIS-{*wmgm,subcort*} and maintains OASIS3 performance.

## 5  Discussion

**Limitations and future work.** The presented work opens up many follow-up questions which will be tackled in future work: (1) Our proposed losses enable better training and performance, but require the tuning of several regularization weights and layer selections which may reveal dataset specific patterns (e.g., rows J, K in Tab. 2). The weights and layers selected here were chosen based on limited exploratory experiments on the validation sets due to computational budgets and future work will exhaustively search the hyperparameter space for optimal performance. (2) Our pretraining assumes accurate non-linear intra-subject registration, which may be non-trivial in edge cases like modalities with strong distortion and artifacts (e.g., eddy corruption in diffusion MRI). However, when studying pre and post-operative imaging (e.g., surgical excision of lesions), large topological changes break the assumptions of our model and will require the development of lesion-masked positive patch sampling methods. (3) We use a two-stage pre-training and finetuning approach and it is plausible that the proposed method can be reduced to a single-stage combined framework. (4) While this paper focused on downstream segmentation finetuning, the pretraining framework is generic to any pixel-level task (e.g., registration) and its extension to such tasks will be explored. (5) While the proposed methods yield strong longitudinal segmentation consistency improvements across all datasets, we note that

Table 2: **Ablation analysis** of our method over loss layers, projection+prediction layer widths (#MLP), loss functions ($\mathcal{L}_{Rec}$, $\mathcal{L}_{cs}$), use of augmentation, and hyperparameters ($\beta, \mu, \gamma$). Mean dice is used for quantification on all datasets. *$\mu = 10^{-3}$ on IBIS-{*wmgm, subcort*} and $\mu = 10^{-2}$ on OASIS3.

| Exp | Loss Layers | #MLP | $\mathcal{L}_{Rec}$ | Aug. | $\beta$ | $\mu$ | $\gamma$ | $\mathcal{L}_{cs}$ | IBIS-*subcort* | IBIS-*wmgm* | OASIS3 |
|---|---|---|---|---|---|---|---|---|---|---|---|
| A | Enc | 256 | | | | | | | 0.829(0.068) | 0.733(0.062) | 0.783(0.16) |
| B | Enc | 2048 | | | | | | | 0.849(0.060) | 0.732(0.073) | 0.809(0.13) |
| C | Enc | 2048 | ✓ | | | | | | 0.859(0.058) | 0.713(0.079) | 0.811(0.13) |
| D | EncDec | 2048 | ✓ | | | | | | 0.860(0.058) | 0.718(0.066) | 0.810(0.13) |
| E | EncDec | 2048 | | ✓ | | | | | 0.858(0.060) | 0.724(0.077) | 0.809(0.13) |
| F | EncDec | 2048 | ✓ | ✓ | | | | | 0.856(0.060) | 0.739(0.067) | 0.812(0.13) |
| G | EncDec | 2048 | ✓ | | 100 | | | | 0.857(0.062) | 0.728(0.074) | 0.809(0.13) |
| H | EncDec | 2048 | ✓ | | | $10^{-3}$ | $10^{-3}$ | | 0.845(0.063) | 0.739(0.074) | 0.804(0.13) |
| I | EncDec | 2048 | ✓ | | 100 | $10^{-3}$ | $10^{-3}$ | | 0.859(0.056) | 0.735(0.061) | 0.811(0.13) |
| J | EncDec | 2048 | ✓ | ✓ | 100 | $10^{-3}$ | $10^{-3}$ | | 0.863(0.057) | 0.758(0.062) | 0.808(0.14) |
| K | EncDec | 2048 | ✓ | ✓ | 100 | $10^{-2}$ | $10^{-3}$ | | 0.853(0.055) | 0.745(0.058) | **0.813(0.13)** |
| L | EncDec | 2048 | ✓ | ✓ | 100 | * | $10^{-3}$ | ✓ | **0.870(0.052)** | **0.806(0.030)** | 0.810(0.13) |

absolute segmentation performance gains in an elderly cohort (OASIS3) are modest in comparison to the rapidly developing infant dataset (IBIS) where higher gains are achieved. Future work will further investigate these performance differences between populations of differing temporal trends. (6) We extended the negative-free framework of [13] to patchwise operations for its relative simplicity and it is plausible that other negative-free similarity terms [6, 20, 62] may further improve results. (7) In our data preparation for pretraining, subject-wise image time-series were registered to a single time-point instead of a subject-specific template, which is known to increase statistical bias [47].

**Limited scope.** The proposed method generically applies to medical image time-series and we do not anticipate negative impacts beyond those that currently exist for segmentation methods. However, while our tasks have impacts on understanding real-world disease mechanisms, we cannot claim any further insight into differences between subpopulations, as such analysis requires close collaboration with clinicians, neuroscientists, and biostatisticians, which is beyond the scope of this work.

**Conclusions.** This paper addressed several open questions regarding the self-supervised pretraining and finetuning of image-to-image architectures on longitudinal volumes using objective functions which exploit both intra-subject spatial and temporal self-similarity. It developed a local negative sample-free framework that trains multiple multi-scale hidden layers of image-to-image architectures that then enabled improved downstream segmentation performance, all while achieving semantically-meaningful representations via careful regularization of the decoder activations. During finetuning, it similarly developed a simple consistency-regularization objective which encourages longitudinal agreement between predictions on unlabeled data. When applied to large-scale neurodeveloping and neurodegerative longitudinal images, the proposed framework yielded improved segmentation performance and temporal consistency, both of which are crucial to statistical analyses of mechanisms of interest such as Alzheimer's Disease (OASIS3) and Autism Spectrum Disorder (IBIS).

## Acknowledgements

The authors are grateful to NIH R01-HD055741-12, R01-MH118362-02S1, 1R01MH118362-01, 1R01HD088125-01A1, R01MH122447, R01-HD059854, U54-HD079124, P50-HD103573, R01ES032294, and the NYS Center for Advanced Technology in Telecommunications (CATT).

Adult longitudinal T1w MRI data were provided by OASIS-3;NIH P50 AG00561, P30 NS09857781, P01 AG026276, P01 AG003991, R01 AG043434, UL1 TR000448, R01 EB009352. Longitudinal developing infant brain T1w/T2w MRI were provided by the Infant Brain Imaging Study (IBIS) Network, which is an NIH-funded Autism Centers of Excellence (ACE) project and consists of a consortium of 10 universities in the U.S. and Canada.

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
