# A    Additional Results and Experiments

**Self-supervised similarity maps** from anatomically-relevant key points are visualized in Figure 5. Using only self-supervision, our model learns semantically and positionally-aware representations.

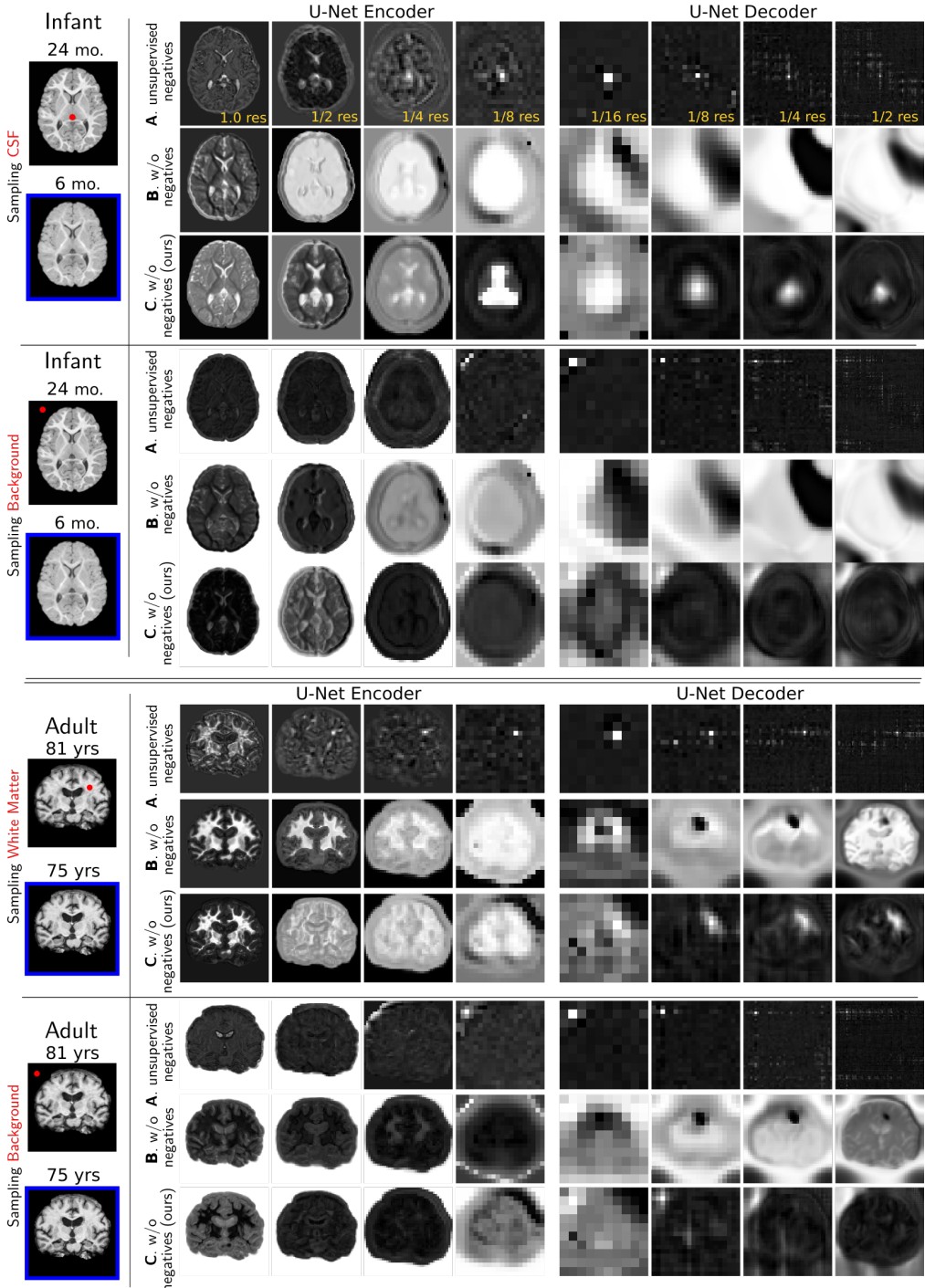

Figure 5: **Self-supervised spatiotemporal multi-scale similarity learning** visualized by sampling a spatial query point at given time point and computing its featurewise similarity to all locations in the key image acquired at a different age. The similarity-maps obtained from both datasets reveal semantically-relevant anatomical representation within the U-Net and projector/predictor networks. This figure follows the same nomenclature (A–C) as Fig. 1 with analysis in Sec. 4 (main text).

**Label-wise longitudinal consistency.** To observe structure-specific performance, we report label-wise longitudinal consistency scores (as measured by STCS [36]) for OASIS3 and IBIS-*subcort* on the test set for the best performing models (as measured by Figure 4 of the main text) in Figure 6. IBIS-*wmgm* was not included as it only has two anatomical structures.

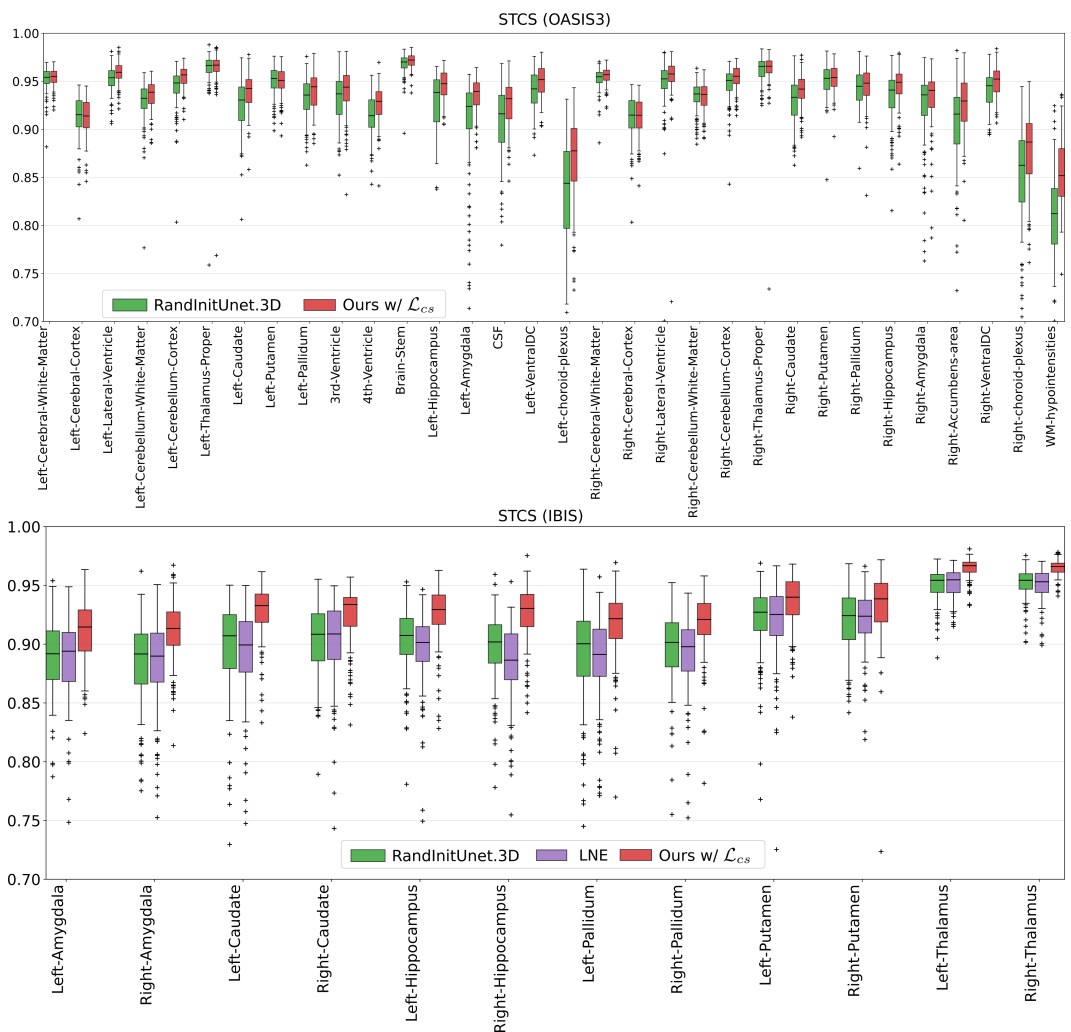

Figure 6: **Per-label longitudinal consistency comparison between the best performing models** as measured by STCS (spatiotemporal consistency of segmentations [36], higher is better) in the one-shot segmentation setting. For space considerations, only the best-performing two and three models are visualized on test set for the OASIS3 and IBIS-*subcort* tasks, respectively.

**Few-shot and fully-supervised segmentation.** While the main body of the paper presents results on one-shot segmentation (corresponding to the common real-world setting of single atlas-based segmentation), the proposed framework has benefits beyond one-shot segmentation. We present results obtained by using 10% and 100% of all the labeled training sets in Tables 3 and 4, respectively. As having more supervised data may necessitate changing our hyperparameters, we explore reducing the weight of the consistency-regularization from $1.0$ to $0.1$ for both experiments. In the 10% setting, we find that Ours w/ $\mathcal{L}_{cs}$ with a weight of $0.1$ obtains optimal performance and consistency. In the 100% setting, we find that including $\mathcal{L}_{cs}$ actually degrades performance and that optimal results are achieved with just the proposed pretraining (Ours w/o $\mathcal{L}_{cs}$). We hypothesize that these trends arise from imperfect deformable registration of the intra-subject images as they were warped considering only their intensity and not semantic structure (see App Section C.1). Therefore, having the imperfect self-supervision of $\mathcal{L}_{cs}$ in the low-annotation regime increases performance. However, hundreds of annotated samples remove the need for $\mathcal{L}_{cs}$, while maintaining the segmentation performance benefits of our pretraining framework over existing work.

Table 3: **_Few_-shot segmentation** using **10%** of the annotated training set for finetuning. This table quantifies test set performance (Dice, IoU, HD95) and longitudinal consistency (ASPC [47], STCS[36]). Mdn: Median.

IBIS-subcort

| Model | Mean(std) Dice | Mdn Dice | Mean(std) IoU | Mdn HD95 | Mdn IoU | ASPC | STCS |
|---|---|---|---|---|---|---|---|
| RandInitUnet.2D | 0.895(0.03) | 0.891 | 0.812(0.06) | 2.390 | 0.804 | 4.618 | 0.909 |
| RandInitUnet.3D | 0.909(0.03) | 0.905 | 0.834(0.05) | 1.819 | 0.827 | 5.531 | 0.911 |
| Context Restore [10] | 0.901(0.03) | 0.898 | 0.821(0.05) | 2.240 | 0.815 | 4.895 | 0.918 |
| LNE [43] | 0.914(0.03) | 0.909 | 0.842(0.05) | 1.845 | 0.834 | 5.807 | 0.911 |
| GLCL [9] | 0.905(0.03) | 0.900 | 0.827(0.05) | 2.486 | 0.819 | 4.369 | 0.913 |
| PCL [63] | 0.904(0.03) | 0.900 | 0.826(0.05) | 2.881 | 0.819 | 4.479 | 0.911 |
| PatchNCE [44] | 0.919(0.03) | 0.915 | 0.851(0.05) | 1.004 | 0.844 | 5.871 | 0.908 |
| Ours w/o $\mathcal{L}_{cs}$ | 0.920(0.03) | 0.916 | 0.853(0.05) | **1.001** | 0.846 | 5.521 | 0.908 |
| Ours w/ 0.1$\mathcal{L}_{cs}$ | **0.923(0.03)** | **0.920** | **0.858(0.04)** | 1.002 | **0.853** | 5.077 | 0.910 |
| Ours w/ 1$\mathcal{L}_{cs}$ | 0.917(0.03) | 0.913 | 0.848(0.05) | 1.002 | 0.840 | **4.655** | **0.925** |

OASIS3

| Model | Mean(std) Dice | Mdn Dice | Mean(std) IoU | Mdn HD95 | Mdn IoU | ASPC | STCS |
|---|---|---|---|---|---|---|---|
| RandInitUnet.2D | 0.852(0.09) | 0.878 | 0.752(0.13) | 1.569 | 0.785 | 4.297 | 0.908 |
| RandInitUnet.3D | 0.867(0.09) | 0.888 | 0.774(0.12) | 1.409 | 0.800 | 2.575 | 0.926 |
| Context Restore [10] | 0.847(0.10) | 0.873 | 0.745(0.13) | 1.660 | 0.777 | 4.528 | 0.916 |
| LNE [43] | 0.872(0.09) | 0.894 | 0.782(0.12) | 1.426 | **0.811** | 2.747 | 0.923 |
| GLCL [9] | 0.861(0.09) | 0.884 | 0.765(0.12) | 1.387 | 0.794 | 4.005 | 0.911 |
| PCL [63] | 0.862(0.09) | 0.885 | 0.766(0.12) | 1.328 | 0.796 | 3.896 | 0.910 |
| PatchNCE [44] | 0.868(0.09) | 0.888 | 0.776(0.12) | 1.347 | 0.800 | 2.666 | 0.927 |
| Ours w/o $\mathcal{L}_{cs}$ | 0.872(0.08) | 0.894 | 0.782(0.12) | 1.372 | 0.810 | 2.647 | 0.923 |
| Ours w/ 0.1$\mathcal{L}_{cs}$ | **0.873(0.08)** | **0.894** | **0.783(0.12)** | **1.335** | 0.809 | 2.602 | 0.926 |
| Ours w/ 1$\mathcal{L}_{cs}$ | 0.868(0.09) | 0.890 | 0.776(0.12) | 1.341 | 0.804 | **2.446** | **0.932** |

Table 4: **_Fully_-supervised segmentation** using **100%** of the annotated training set for finetuning. This table quantifies test set performance (Dice, IoU, HD95) and longitudinal consistency (ASPC [47], STCS[36]). Mdn: Median.

IBIS-subcort

| Model | Mean(std) Dice | Mdn Dice | Mean(std) IoU | Mdn HD95 | Mdn IoU | ASPC | STCS |
|---|---|---|---|---|---|---|---|
| RandInitUnet.2D | 0.902(0.03) | 0.898 | 0.824(0.05) | 2.634 | 0.815 | 4.308 | 0.917 |
| RandInitUnet.3D | 0.922(0.03) | 0.917 | 0.856(0.04) | 1.730 | 0.847 | 5.517 | 0.916 |
| Context Restore [10] | 0.909(0.03) | 0.906 | 0.835(0.05) | 1.011 | 0.828 | 4.125 | 0.922 |
| LNE [43] | 0.928(0.02) | 0.924 | 0.867(0.04) | 1.000 | 0.859 | 5.253 | 0.911 |
| GLCL [9] | 0.914(0.03) | 0.909 | 0.843(0.05) | 1.011 | 0.834 | 4.191 | 0.917 |
| PCL [63] | 0.913(0.03) | 0.909 | 0.842(0.05) | 1.006 | 0.833 | **4.122** | 0.918 |
| PatchNCE [44] | 0.928(0.02) | 0.924 | 0.867(0.04) | 1.001 | 0.860 | 5.470 | 0.910 |
| Ours w/o $\mathcal{L}_{cs}$ | **0.933(0.02)** | **0.930** | **0.876(0.04)** | 1.000 | **0.870** | 6.666 | 0.901 |
| Ours w/ 0.1$\mathcal{L}_{cs}$ | 0.930(0.02) | 0.926 | 0.870(0.04) | 1.000 | 0.863 | 5.298 | 0.910 |
| Ours w/ 1$\mathcal{L}_{cs}$ | 0.922(0.02) | 0.917 | 0.856(0.04) | 1.000 | 0.846 | 4.605 | **0.926** |

OASIS3

| Model | Mean(std) Dice | Mdn Dice | Mean(std) IoU | Mdn HD95 | Mdn IoU | ASPC | STCS |
|---|---|---|---|---|---|---|---|
| RandInitUnet.2D | 0.844(0.10) | 0.874 | 0.741(0.13) | 1.632 | 0.777 | 4.438 | 0.914 |
| RandInitUnet.3D | 0.878(0.08) | 0.898 | 0.791(0.11) | 1.371 | 0.817 | 5.570 | 0.920 |
| Context Restore [10] | 0.864(0.09) | 0.886 | 0.769(0.12) | 1.415 | 0.797 | 4.202 | 0.927 |
| LNE [43] | 0.882(0.08) | 0.904 | 0.796(0.11) | 1.280 | 0.825 | 4.140 | 0.926 |
| GLCL [9] | 0.864(0.09) | 0.887 | 0.769(0.12) | 1.424 | 0.799 | 3.590 | 0.926 |
| PCL [63] | 0.865(0.09) | 0.890 | 0.771(0.12) | 1.315 | 0.802 | 3.328 | 0.926 |
| PatchNCE [44] | 0.869(0.08) | 0.889 | 0.777(0.11) | 1.328 | 0.801 | 2.599 | 0.937 |
| Ours w/o $\mathcal{L}_{cs}$ | **0.885(0.08)** | **0.907** | **0.801(0.11)** | 1.246 | **0.831** | 2.919 | 0.933 |
| Ours w/ 0.1$\mathcal{L}_{cs}$ | 0.882(0.08) | 0.904 | 0.796(0.11) | **1.233** | 0.825 | 2.647 | 0.934 |
| Ours w/ 1$\mathcal{L}_{cs}$ | 0.877(0.08) | 0.898 | 0.789(0.11) | 1.245 | 0.817 | **2.368** | **0.939** |

**U-Net configuration.** For consistency, we use the same base U-Net architecture for all baselines. Its configuration is modeled based on the one-shot segmentation mean Dice validation results on OASIS3 presented in Table 5 and evaluated over several training crop sizes, normalization layers, and channel width multipliers. Importantly, all configurations were trained without any pretraining to obtain a baseline. We observe that a random crop window of $128^3$ is optimal (rows A, B, C) and find that Instance Normalization (row D) instead of Batch Normalization degrades performance. Finally, we observe almost no performance degradation when reducing the model size via the channel width multiplier from 24 to 16 (row C vs E) and therefore use 16 for computational efficiency.

Table 5: **Base network tuning.** Randomly initialized UNet performance over various crop sizes, normalization types, and the # channels of the first convolution layer. E is the final configuration for the base network used in all baselines as it best trades-off memory usage and performance.

| Exp | Crop size | Normalization | Channel width | mean(std) dice |
|---|---|---|---|---|
| A | $64 \times 64 \times 64$ | batch norm | 24 | 0.768 ($\pm$ 0.14) |
| B | $128 \times 128 \times 128$ | batch norm | 24 | **0.795 ($\pm$ 0.14)** |
| C | $160 \times 160 \times 192$ | batch norm | 24 | 0.786 ($\pm$ 0.14) |
| D | $128 \times 128 \times 128$ | instance norm | 24 | 0.777 ($\pm$ 0.14) |
| E | $128 \times 128 \times 128$ | batch norm | 16 | 0.792 ($\pm$ 0.15) |
| F | $128 \times 128 \times 128$ | batch norm | 8 | 0.786 ($\pm$ 0.14) |

**Training a randomly-initialized U-Net with $\mathcal{L}_{cs}$.** To investigate the standalone benefit of $\mathcal{L}_{cs}$ without any regularized pretraining, we apply $\mathcal{L}_{cs}$ to a randomly initialized 3D U-Net in the one-shot segmentation setting for IBIS-*subcort* and report its results in Fig. 7.

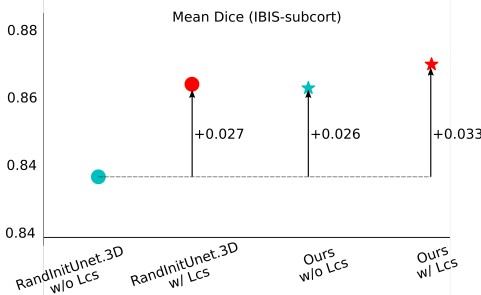

Figure 7: Stand-alone combination of the proposed $\mathcal{L}_{cs}$ consistency-regularization and one-shot segmentation training of a randomly-initialized U-Net. In the one-shot setting, $\mathcal{L}_{cs}$ improves mean dice by 0.027, which is comparable to the proposed model without $\mathcal{L}_{cs}$. Optimal results are obtained when combined with our regularized pretraining framework.

**Using last decoder layer in loss.** We investigate the impact of including the full-resolution feature (layer 21 in Table 9) in the `EncDec` loss configuration described in Section B.1. As shown in Figure 8, this notably degrades performance, indicating sensitivity to the exact layers used for $\mathcal{L}_{sim}$ calculation.

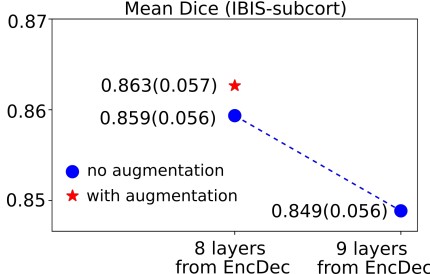

Figure 8: A comparison between using different number of layers during pretraining. We used '8 layers from EncDec' as our final layer selection as a trade-off between memory and performance.

**Additional modeling decisions.** In addition to the quantitative ablations over our modeling choices included in the main paper and appendix, we make a few modeling decisions detailed in Table 6 based on qualitative assessments of the representation quality using feature similarity maps.

Table 6: Additional modeling decisions. The final selections are bolded.

| Parameter | Search space |
|---|---|
| # of layers in projector | 2,**3** |
| non-linearity | ELU, **ReLU**, LeakyReLU |
| within subject registration | affine only, **affine and deformable** |
| feature selection for similarity loss | **conv output**, activation output |

**Additional one-shot quantification.** As Table 1 and Figure 4 (main text) report median scores, Table 7 reports means and standard deviations for further interpretation. Similar to the medians, our method improves both mean performance and longitudinal consistency on IBIS-{*wmgm, subcort*}. On OASIS3, our Dice and IoU performance improves on all baselines except for LNE (with whom it is strongly competitive) and exceeds all baselines in terms of longitudinal consistency (ASPC, STCS).

Table 7: **One-shot segmentation performance** to complement results presented in Tab. 1 and Fig. 4 of the main text, using means instead of medians. *In OASIS3, we exclude the `Right Accumbens Area` and `4th Ventricle` labels when calculating mean HD95 due to numerical issues for some of the baselines (`RandInitUnet.2D`, `Context Restore`, and `PCL`).

| | | | IBIS-subcort | | |
|---|---|---|---|---|---|
| Model | Dice | IoU | HD95 | ASPC | STCS |
| RandInitUnet.2D | 0.828(0.07) | 0.712(0.10) | 4.784(6.32) | 7.08(7.85) | 0.893(0.03) |
| RandInitUnet.3D | 0.836(0.06) | 0.724(0.09) | 3.603(5.27) | 5.29(5.12) | 0.908(0.02) |
| Context Restore [10] | 0.838(0.07) | 0.727(0.09) | 5.196(9.27) | 8.22(9.16) | 0.893(0.02) |
| LNE [43] | 0.847(0.06) | 0.739(0.09) | 3.478(5.33) | 5.83(5.41) | 0.905(0.02) |
| GLCL [9] | 0.835(0.07) | 0.722(0.10) | 3.640(4.16) | 5.47(6.23) | 0.905(0.03) |
| PCL [63] | 0.833(0.07) | 0.720(0.09) | 3.528(3.56) | 5.78(6.10) | 0.902(0.03) |
| PatchNCE [44] | 0.856(0.06) | 0.753(0.09) | 1.482(1.75) | 5.93(5.79) | 0.909(0.02) |
| Ours w/o $\mathcal{L}_{cs}$ | 0.863(0.06) | 0.763(0.08) | 1.510(2.62) | 5.68(6.00) | 0.909(0.02) |
| Ours w/ $\mathcal{L}_{cs}$ | **0.870(0.05)** | **0.773(0.08)** | **1.408(1.24)** | **4.26(4.32)** | **0.929(0.02)** |

| | | | IBIS-wmgm | | |
|---|---|---|---|---|---|
| Model | Dice | IoU | HD95 | ASPC | STCS |
| RandInitUnet.2D | 0.672(0.07) | 0.510(0.08) | 3.274(1.00) | 11.51(11.79) | 0.832(0.03) |
| RandInitUnet.3D | 0.713(0.08) | 0.560(0.09) | 3.788(0.70) | 5.52(5.48) | 0.863(0.05) |
| Context Restore [10] | 0.590(0.19) | 0.444(0.19) | 8.273(3.56) | 29.24(26.76) | 0.813(0.10) |
| LNE [43] | 0.716(0.07) | 0.563(0.09) | **3.201(0.98)** | 5.35(6.34) | 0.858(0.04) |
| GLCL [9] | 0.707(0.05) | 0.550(0.06) | 4.112(1.14) | 8.47(8.85) | 0.858(0.02) |
| PCL [63] | 0.715(0.08) | 0.562(0.09) | 4.974(1.75) | 10.65(13.29) | 0.868(0.05) |
| PatchNCE [44] | 0.753(0.05) | 0.607(0.06) | 4.344(0.66) | 3.78(4.18) | 0.889(0.03) |
| Ours w/o $\mathcal{L}_{cs}$ | 0.758(0.06) | 0.614(0.08) | 3.291(0.66) | **2.46(2.74)** | 0.882(0.03) |
| Ours w/ $\mathcal{L}_{cs}$ | **0.806(0.03)** | **0.676(0.04)** | 3.237(0.47) | 4.15(3.32) | **0.910(0.02)** |

| | | | OASIS3 | | |
|---|---|---|---|---|---|
| Model | Dice | IoU | HD95* | ASPC | STCS |
| RandInitUnet.2D | 0.781(0.15) | 0.661(0.17) | 3.425(5.00) | 10.27(28.84) | 0.873(0.05) |
| RandInitUnet.3D | 0.804(0.13) | 0.689(0.16) | 3.424(5.85) | 3.53(6.35) | 0.923(0.05) |
| Context Restore [10] | 0.801(0.14) | 0.687(0.16) | 4.841(10.00) | 6.88(15.21) | 0.899(0.04) |
| LNE [43] | **0.813(0.13)** | 0.700(0.15) | **3.171(5.50)** | 8.81(31.94) | 0.898(0.04) |
| GLCL [9] | 0.792(0.14) | 0.674(0.16) | 3.330(5.25) | 5.30(11.12) | 0.903(0.05) |
| PCL [63] | 0.796(0.14) | 0.680(0.17) | 3.339(5.41) | 5.69(13.40) | 0.899(0.05) |
| PatchNCE [44] | 0.809(0.13) | 0.697(0.16) | 3.298(5.93) | 3.49(6.70) | 0.920(0.04) |
| Ours w/o $\mathcal{L}_{cs}$ | **0.813(0.13)** | **0.702(0.15)** | 3.289(5.68) | 3.40(6.11) | 0.923(0.04) |
| Ours w/ $\mathcal{L}_{cs}$ | 0.810(0.13) | 0.697(0.16) | 3.733(6.41) | **2.76(4.96)** | **0.934(0.03)** |

**Additional IBIS-*wmgm* segmentation results.** For one-shot IBIS-*wmgm* segmentation, we train on a single 36 month old T1w/T2w MR image and quantitatively evaluate on 6 month old MR images (for additional motivation see Appendix Section D.2). In Figure 9, we visualize predictions using additional baselines and our framework on a held-out test subject. As the infant brain matures over time (Appendix Section D.2), the rows are arranged in order of the most to the least amount of biological domain-shift w.r.t. the 36 month-old training image. Further, as the 12, 24, and 36 month old held-out images do not have ground truth segmentations available, we are restricted to qualitative evaluations of segmentation quality.

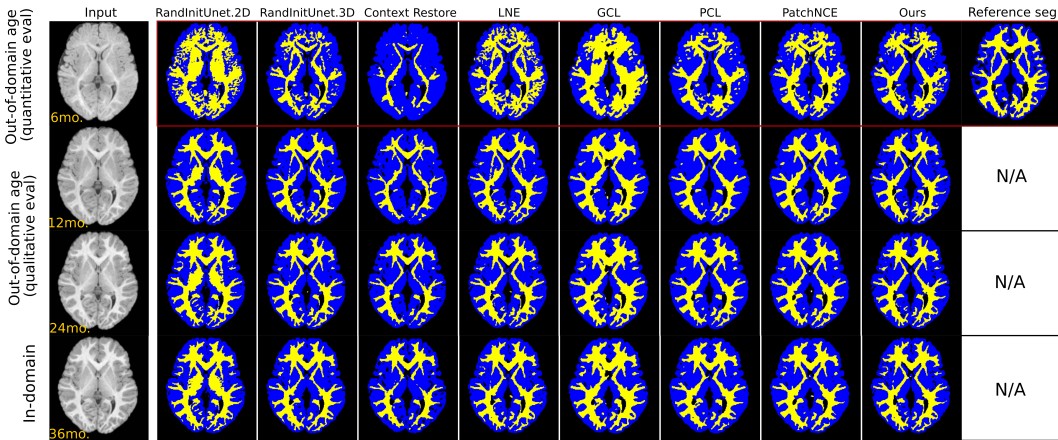

Figure 9: Infant brain segmentation on a held-out subject under the strong domain shift induced by training on a single 36 month-old image and testing on the remaining timepoints. Under the most extreme domain shift of row 1, we see that all baselines yield highly-inaccurate predictions, while our method yields higher segmentation quality.

# B  Additional Implementation Details

## B.1  Additional Training Details

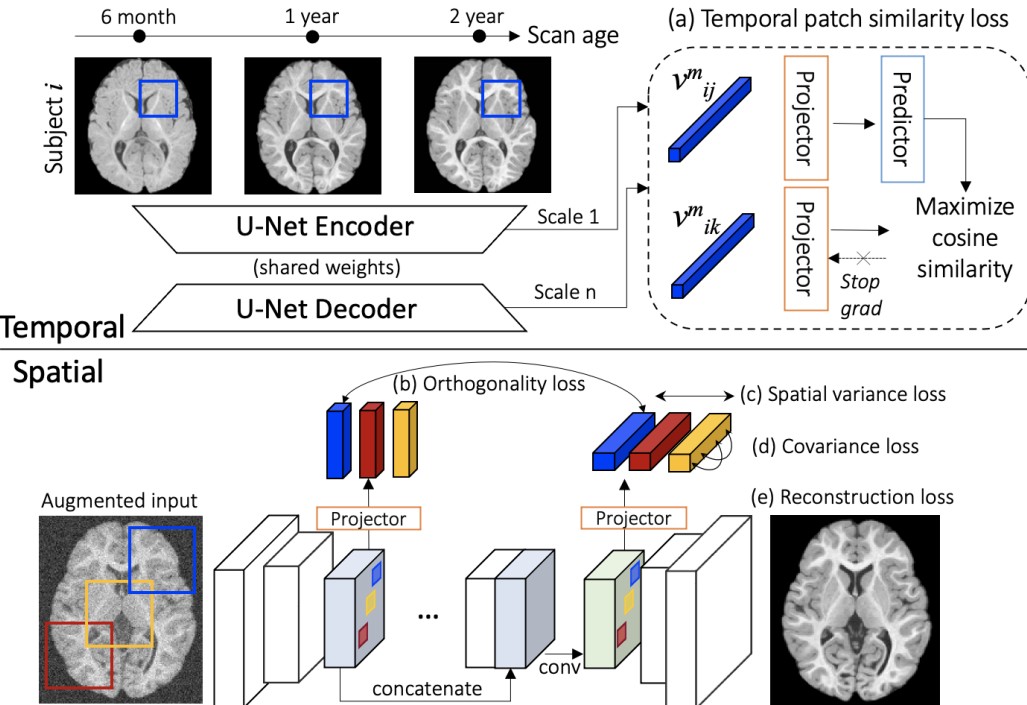

Figure 10: **Pre-training Overview.** A visual illustration of the proposed losses (a, b, c, d, e) for spatiotemporal representation learning with a U-Net. An overview of (a)–(d) is in main text Fig.1.

Figure 10 illustrates an in-depth overview of the proposed pretraining and representation learning framework. In the proposed loss configuration (`EncDec`), Pre-activation features from layers $\{1, 3, 5, 7, 9, 12, 15, 18\}$ in Table 5 are sampled spatially to extract channel-wise vectors which are fed into the projector and predictor MLPs for $\mathcal{L}_{sim}$ calculation. Patch features from layer $\{8, 12\}$ in the bottleneck (processed by the projector) are used for the orthogonality regularizer $\mathcal{L}_O$ and layers $\{12, 15, 18\}$ in the decoder are used for the $\mathcal{L}_S$ and $\mathcal{L}_C$ regularizers. In the ablations in Table 2 (main text), the `Enc` loss setting refers to using features from layers $\{1, 2, 3, 4, 5, 6, 7, 8, 9, 10, 11\}$ for $\mathcal{L}_{sim}$.

Each iteration of pretraining loads three corresponding crops from two intra-subject images each for $\mathcal{L}_{sim}$ calculation. Higher batch sizes could have been used for the proposed loss functions but were not for consistency with one of our baselines (`PatchNCE` [44]) which is highly memory-intensive due to the need for a large number of negative samples alongside 3D computation.

During finetuning, the segmentation loss ($\mathcal{L}_{sup}$) is applied to the original unwarped images and the consistency loss ($\mathcal{L}_{cs}$) is applied to the nonlinearly aligned images. When calculating $\mathcal{L}_{sup}$ and $\mathcal{L}_{cs}$, we perform individual forward/backward passes for each loss due to memory limitations as gradient accumulation over two forward passes was found to decrease performance on a validation set.

## B.2  Architectures

The MLP architectures for the **projector** and **predictor** networks used for pretraining and representation learning are given in Table 8. The **U-Net** architecture used for both pretraining and finetuning is given in Table 9. During pretraining, an additional convolutional layer is used following layer 23 to reconstruct/denoise the input. During finetuning, a channel-wise softmax layer is attached following layer 23 for segmentation.

Table 8: Projector and predictor MLP architectures.

Projector (f)

| id | Layer | Output size |
|----|-------|-------------|
| 0 | FC(2048), BN, ReLU | (num patches $\times$ batch size) $\times$ 2048 |
| 1 | FC(2048), BN, ReLU | (num patches $\times$ batch size) $\times$ 2048 |
| 2 | FC(2048), BN | (num patches $\times$ batch size) $\times$ 2048 |

Predictor (p)

| id | Layer | Output size |
|----|-------|-------------|
| 0 | FC(256), BN, ReLU | (num patches $\times$ batch size) $\times$ 256 |
| 1 | FC(2048), BN, ReLU | (num patches $\times$ batch size) $\times$ 2048 |

Table 9: U-Net architecture. All convolutional layers use $3 \times 3$ kernels. BN: Batch Normalization (using default PyTorch momentum). The batch size dimension is denoted $bs$ and $nc$ is the starting channel width multiplier of the model. We choose $nc = 16$ consistently throughout all models based on Table5. $n$ indicates the number of output channels and is set to the number of labels.

| id | Layer | Output size |
|----|-------|-------------|
| 0 | Conv3D(nc), BN, ReLU | bs, w, h, d, nc |
| 1 | Conv3D(nc), BN, ReLU | bs, w, h, d, nc |
| 2 | Conv3D(nc), BN, ReLU | bs, w, h, d, nc |
| 3 | MaxPool(2), Conv3D(2nc), BN, ReLU | bs, w/2, h/2, d/2, 2nc |
| 4 | Conv3D(2nc), BN, ReLU | bs, w/2, h/2, d/2, 2nc |
| 5 | MaxPool(2), Conv3D(4nc), BN, ReLU | bs, w/4, h/4, d/4, 4nc |
| 6 | Conv3D(4nc), BN, ReLU | bs, w/4, h/4, d/4, 4nc |
| 7 | MaxPool(2), Conv3D(8nc), BN, ReLU | bs, w/8, h/8, d/8, 8nc |
| 8 | Conv3D(8nc), BN, ReLU | bs, w/8, h/8, d/8, 8nc |
| 9 | MaxPool(2), Conv3D(16nc), BN, ReLU | bs, w/16, h/16, d/16, 16nc |
| 10 | Conv3D(16nc), BN, ReLU | bs, w/16, h/16, d/16, 16nc |
| 11 | Upsample(2), Concatenate with 8 | bs, w/8, h/8, d/8, 24nc |
| 12 | Conv3D(16nc), BN, ReLU | bs, w/8, h/8, d/8, 8nc |
| 13 | Conv3D(16nc), BN, ReLU | bs, w/8, h/8, d/8, 8nc |
| 14 | Upsample(2), Concatenate with 6 | bs, w/4, h/4, d/4, 12nc |
| 15 | Conv3D(4nc), BN, ReLU | bs, w/4, h/4, d/4, 4nc |
| 16 | Conv3D(4nc), BN, ReLU | bs, w/4, h/4, d/4, 4nc |
| 17 | Upsample(2), Concatenate with 4 | bs, w/2, h/2, d/2, 6nc |
| 18 | Conv3D(2nc), BN, ReLU | bs, w/2, h/2, d/2, 2nc |
| 19 | Conv3D(2nc), BN, ReLU | bs, w/2, h/2, d/2, 2nc |
| 20 | Upsample(2), Concatenate with 2 | bs, w, h, d, 3nc |
| 21 | Conv3D(nc), BN, ReLU | bs, w, h, d, nc |
| 22 | Conv3D(nc), BN, ReLU | bs, w, h, d, nc |
| 23 | Conv3D(n) | bs, w, h, d, n |

### B.3 Baseline Reimplementation Details

**LNE.** LNE [43] is a self-supervised global representation learning method that models longitudinal effects in an autoencoder latent space and is repurposed here for longitudinal segmentation. During pretraining, we require the same base U-Net architecture for consistency with other baselines. Therefore, to extract a 1024D global representation as in the original paper, we add a global pooling layer on the output of layer 13 in Table 9, followed by a single 1024D fully-connected layer. With respect to their hyperparameters, due to the larger base network size, we reduce the batch size from 64 to 16 images due to memory considerations. We maintain the other hyperparameters of LNE including downsampling the images to $64^3$, using $N = 5$ neighborhoods, and weighing auxiliary losses as $\lambda_{dir} = 1$ and $\lambda_{rec} = 2$ (see original work for definitions [43]).

**PatchNCE.** PatchNCE [44] proposes a framework for unsupervised multi-scale patchwise contrastive learning applied to the problem of unpaired image translation and is extended to serve as a baseline in this work as it shares a partially similar motivation. To adapt it to our generic longitudinal representation learning setting, we impose the same modeling assumption as our method (spatial indices in correspondence across an image time-series are positive samples and all other spatial indices are negatives) and modify both its data sampling and loss calculation. First, we perform forward passes on $t$ corresponding intra-subject crops. At each selected layer, $N$ feature vectors are randomly sampled for the longitudinal InfoNCE [32] loss: $\mathcal{L} = \sum_i \frac{1}{|P(i)|} \sum_{p \in P(i)} - \log \frac{e^{z_i \cdot z_p / \tau}}{\sum_{s \in S(i)} e^{z_i \cdot z_s / \tau}}$, where for any query feature index $i$, $P(i)$ is the set of indices of all positives (from the same subject at the same spatial index) and $S(i)$ is the set of all query indices (with size $(t \times N) - 1$, where $t$ is the number of timepoints in the batch and $N$ is the number of patches). We choose $t = 3$ timepoints and $N = 768$ patches in our experiments due to memory limits. An ablation study of PatchNCE over the layers used for the loss function and the temperature is given in Table 10. For prototyping efficiency, we perform baseline tuning using a three-layer MLP of size 256 and use validation Dice as the model selection criterion to choose $F$ as our final configuration. For fair comparison with our model, we increase the MLP width in the `PathNCE` model to 2048 in all other experiments reported, although this does not significantly alter results.

**Context Restoration**. [10] proposes context restoration as a pretext task for self-supervised representation learning which consists of randomly and repeatedly swapping image patches and training the network to restore the original image. We extend Algorithm 1 in [10] to use 3D inputs and targets and tune it with a varying number of swapping iterations (#swaps) and swapped patch sizes. Downstream validation dice scores are provided in Table 11 and configuration $D$ is chosen as the final model.

**PCL, GLCL, and 2D U-Net.** PCL and GLCL [9, 63] are 2D slice-based contrastive learning methods designed for 3D volume segmentation. We follow the optimal configurations (e.g. temperature, number of partitions, etc.) reported in the original papers. As these methods use a 2D U-Net trained on 2D slices, we additionally benchmark against a randomly initialized 2D U-Net to obtain a baseline for 2D methods. We use a batch size of 128 for all 2D baselines. Within each batch, we compare two different data sampling strategies: (1) randomly sampling 2D slices across all 3D volumes; (2) sampling intra-volume slices within each batch (i.e. treat one of the dimensions from a $128 \times 128 \times 128$ crop as the batch dimension for 2D networks). We train a randomly initialized 2D U-Net for the one-shot segmentation task, and found that (2) significantly increases the validation mean Dice over (1) on OASIS3 from $0.708(0.193)$ to $0.780(0.137)$ under the same architecture and optimization set up. We therefore use batch sampling strategy (2) for all 2D model benchmarks (PCL, GLCL, RandInitUnet.2D).

Table 10: Parameter search (on IBIS-subcort) of patchNCE pretraining over layers included for multiscale patchNCE loss, and temperature. F is used for the final comparison.

| Exp | layers | temperature | Mean(std) dice |
|-----|--------|-------------|----------------|
| A | Enc | 0.07 | 0.834 ($\pm$ 0.060) |
| B | Enc | 0.1 | 0.834 ($\pm$ 0.062) |
| C | Enc | 0.2 | 0.823 ($\pm$ 0.067) |
| D | Enc | 0.5 | 0.830 ($\pm$ 0.064) |
| E | EncDec | 0.07 | 0.854 ($\pm$ 0.056) |
| F | EncDec | 0.1 | **0.856 ($\pm$ 0.054)** |
| G | EncDec | 0.2 | 0.846 ($\pm$ 0.058) |
| H | EncDec | 0.5 | 0.850 ($\pm$ 0.054) |

Table 11: Context restoration parameter search over the size of local patches and the number of patch swap iterations on OASIS3. Validation mean(std) dice is used for selecting the configuration D.

| Exp | patch size | # swaps | Mean(std) Dice |
|-----|-----------|---------|----------------|
| A | $16^3$ | 10 | 0.778 ($\pm$ 0.141) |
| B | $24^3$ | 10 | 0.779 ($\pm$ 0.142) |
| C | $16^3$ | 30 | 0.775 ($\pm$ 0.138) |
| D | $16^3$ | 50 | **0.782 ($\pm$ 0.135)** |
| E | $16^3$ | 100 | 0.778 ($\pm$ 0.160) |

# C  Additional data preparation details

## C.1  Registration

### C.1.1  Affine registration

To warp all images to a common space, we warp all images affinely to a constructed template [4] (with an affine transformation model) using `ANTs` [1] with the following command:

```
antsMultivariateTemplateConstruction2.sh \
-d 3 \
-o OUTPUT_FOLDER/T \
-i 1 -g 0.2 -j 128 -c 2 -r 1 -n 0 -m MI -l 1 \
-t Affine INPUT_FOLDER/*t1.nii.gz'
```

Once affinely aligned on a dataset-wide level, we proceed with longitudinal intra-subject deformable alignment for the calculation of $\mathcal{L}_{sim}$ and $\mathcal{L}_{cs}$, as described below.

### C.1.2  Nonlinear Deformable Registration

Proper spatial correspondence of positive samples for $\mathcal{L}_{sim}$ and label maps for $\mathcal{L}_{cs}$ (the similarity and consistency losses, respectively) requires nonlinear/deformable registration of all intra-subject images to a common reference. To this end, we employ the `ANTs SyN` algorithm from [3] to register within-subject images to a single timepoint in a series of acquisitions with the following command:

```
fix = INPUT_FOLDER/{subj}_{trg_tp}_t1.nii.gz
for src_tp in tps:
    moving = INPUT_FOLDER/{subj}_{src_tp}_t1.nii.gz
    antsRegistration \
    --verbose 1 \
    --dimensionality 3 \
    --float 1 \
    --output [OUTPUT_FOLDER/{subj}_{src_tp}_to_{trg_tp}_t1_, \
             OUTPUT_FOLDER/{subj}_{src_tp}_to_{trg_tp}_t1_Warped.nii.gz, \
             OUTPUT_FOLDER/{subj}_{src_tp}_to_{trg_tp}_t1_InvWarped.nii.gz] \
    --transform SyN[0.15, 9, 0.2] \
    --metric CC[{fix}, {moving}, 1, 2, Random, 0.4] \
    --convergence [250x125x50, 1e-5, 10] \
    --shrink-factors 4x2x1 \
    --smoothing-sigmas 2x1x0vox \
    --interpolation Linear
```

where `fix` is the intra-subject image from a selected timepoint `trg_tp` to which all other timepoints `src_tp` are registered.

## C.2  Additional preprocessing and label generation

With respect to OASIS3, the publicly-available dataset arrives preprocessed with cross-sectional FreeSurfer processing (intensity and geometric normalization and segmentation). As cross-sectional FreeSurfer does not account for longitudinal effects [47], FreeSurfer V6.0 is used for additional longitudinal processing on top of the publicly-released cross-sectional FreeSurfer segmentations. A comparison between longitudinal and cross-sectional FreeSurfer analysis based on training set is given in Fig. 11, where we observe notable improvements to longitudinal segmentation consistency (as quantified by STCS [36]) over all structures.

For IBIS-{*subcort, wmgm*}, all T1w/T2w MR images are preprocessed following standard procedures described in [50] of gradient distortion correction, bias-field correction, within-subject and within time-point multi-modality registration, and brain extraction. For IBIS-*subcort*, ground truth label generation follows the segmentation procedures outlined in [50]. For IBIS-*wmgm*, we use

---

[1] `https://github.com/ANTsX/ANTs`

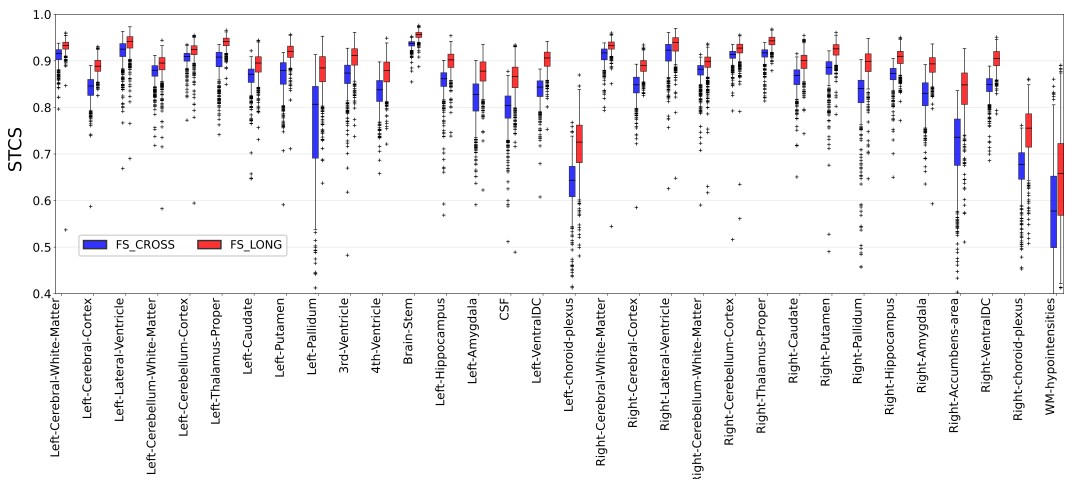

Figure 11: Longitudinal FreeSurfer processing (for anatomical segmentation used as ground-truth targets) leads to substantial improvements over the publicly-released cross-sectional FreeSurfer processing of OASIS3 in longitudinal-consistency metrics such as STCS [36] (higher is better). See section C.2 for further motivation.

segmentations generated by a fully-supervised network trained on an external dataset, see Appendix Section D.2 for more details.

Once preprocessed and aligned with methods described in Section C.1, the images are cropped to a common field-of-view ($128 \times 192 \times 160$ for IBIS, $160 \times 192 \times 160$ for OASIS3, corresponding to smaller brain volumes for infants vs. adults) for compatibility with common multi-resolution neural network architectures.

## C.3 Data splitting with repeat acquisitions

We perform a 70-10-20 train, validation, and test split on subject-wise basis from a total of $N$ subjects and $M$ images ($M > N$ due to image time-series acquisitions), with the entire procedure illustrated in Figure 12. This level of the data split hierarchy ($N\_cross$ subjects and $M\_cross$ images) is used for segmentation training and evaluation. For longitudinal pretraining and longitudinal consistency evaluation (ASPC, STCS in Table 1 and Figure 4 of the main text), we further filter $N\_long$ subjects with at least 2 acquisitions per subject to obtain $M\_long$ images.

Finally, IBIS-*subcort* and IBIS-*wmgm* are segmentation tasks which share the same unlabeled pretraining data. Importantly, we note that as IBIS-*wmgm* is evaluated and tested only on 6-month-old MR images (representing the strongest domain shift, see Appendix Section D.2 for its motivation), its segmentation validation and test sets contain fewer images than IBIS-*subcort*. Table 12 provides the exact sample sizes for each task.

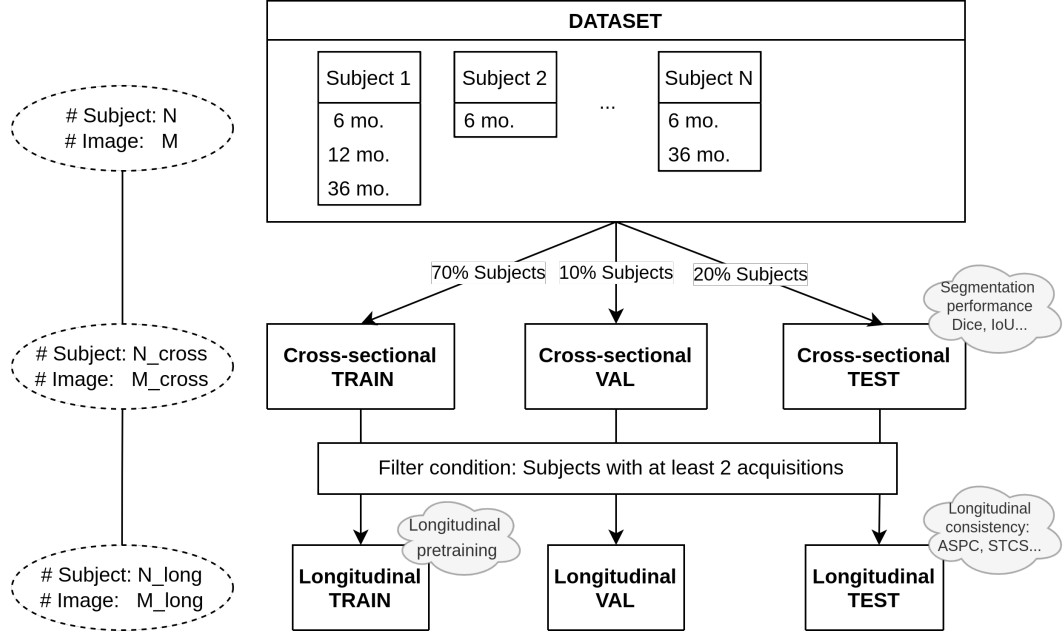

Figure 12: **Longitudinal data selection and splitting criteria. Row 1 → 2:** Given $M$ images from $N$ subjects, we apply a 70-10-20 train/val/test *subject-wise* split to avoid data leakage. This level of data split hierarchy is compatible with the calculation of segmentation performance scores which are agnostic to repeat acquisitions such as the Dice coefficient and mean Intersection-over-Union. **Row 2 → 3:** For longitudinal pretraining and calculation of longitudinal consistency scores (such as ASPC and STCS), we further filter the data splits to only contain subjects with two or more acquisitions. Actual dataset-specific numbers reported in Table 12.

Table 12: **Dataset-specific sample sizes of images (N) and subjects (M).** Table best interpreted in conjunction with Fig. 12. Abbreviations include *_long*: longitudinal and *_cross*: cross-sectional.

|  | IBIS | | | |
| --- | --- | --- | --- | --- |
|  | IBIS-subcort | | IBIS-wmgm | |
| Split | (N_cross, M_cross) | (N_long, M_long) | (N_cross, M_cross) | (N_long, M_long) |
| Total | (552, 1272) | (455, 1175) | (552, 1272) | (552, 1272) |
| Train | (386, 887) | (313, 814) | (386, 887) | (313, 814) |
| Validation | (55, 133) | (50, 128) | (55, 42) | (50, 128) |
| Test | (111, 252) | (92, 233) | (111, 80) | (92, 233) |

|  | OASIS3 | |
| --- | --- | --- |
| Split | (N_cross, M_cross) | (N_long, M_long) |
| Total | (992, 1639) | (422, 1069) |
| Train | (694, 1147) | (293, 746) |
| Validation | (98, 166) | (48, 116) |
| Test | (200, 326) | (81, 207) |

# D  Miscellaneous Details

## D.1  Data availability and IRB information

**IBIS.** The IBIS/Autism MRI data is available through NIH NDA[2]. The Infant Brain Imaging Study (IBIS) Network is a National Institutes of Health–funded Autism Center of Excellence project and consists of a consortium of eight universities in the United States and Canada. Parents provided informed consent and the institutional review board at each site approved the research protocol.

**OASIS-3.** OASIS-3 [34] is publicly available through the OASIS webpage[3]. Ethical approval was obtained by the relevant ethics committees and informed consent was obtained from all participants following procedures set by the IRB at the Washington University School of Medicine.

## D.2  IBIS-wmgm segmentation task details

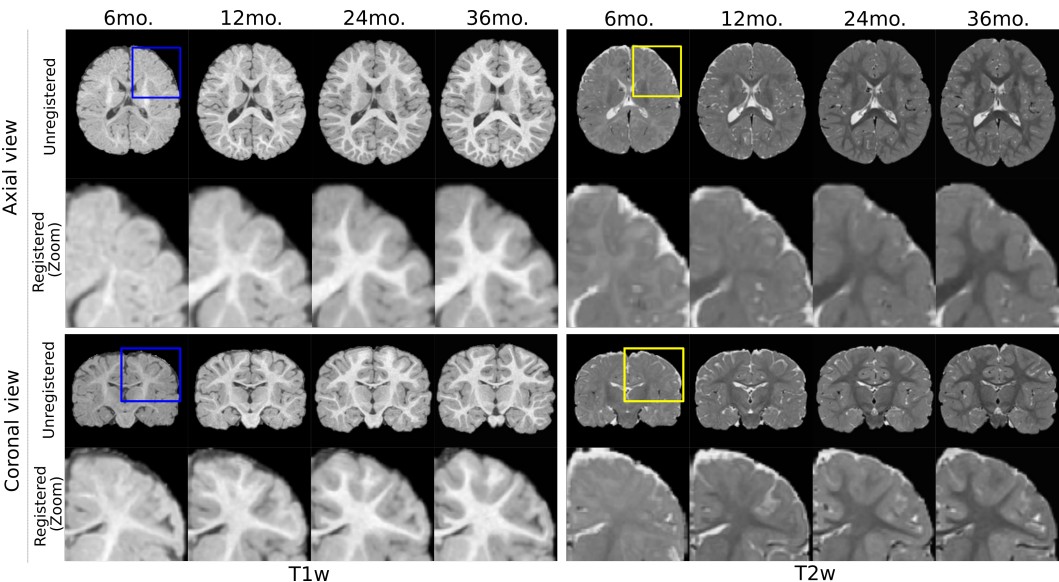

Figure 13: A showcase of **brain appearance maturation** through the early lifespan. Left: T1w; right: T2w images of developing infant brains from axial/coronal view.

**Infant white matter maturation.** Neurodevelopment in infants and toddlers is a highly-complex process of macro and micro-structural changes (for example, growth and myelination, respectively). Relevant to the scope of this paper, real-world tissue segmentation (into grey and white matter) of infant brains at approximately 6 months of age (post-birth) is highly difficult due to ongoing white matter myelination giving the brain an "isointense" (flat in intensity) appearance which ambiguates anatomical edges. Fig. 13 visualizes this phenomenon.

Currently, existing datasets for supervised segmentation of isointense infant brains [53] construct their ground truth labels by first nonlinearly registering algorithmic segmentations of an older timepoint from a given subject image time-series (which is straightforward to segment with existing tools) and then manually correcting the warped labels with the expertise of neuroradiologists.

In our one-shot segmentation setting, we take a partially analogous approach to segmentation of isointense infant brains. We first algorithmically segment [46] a single arbitrarily-selected 36 month old T1w/T2w MR image and use this reliable segmentation to train all segmentation baselines and the proposed method. These methods which have only been trained on a 36 month old brain are then quantitatively evaluated on isointense 6 month old brain tissue segmentation in this paper. The target labels for the 6 month old evaluation set are generated algorithmically with supervised segmentation networks following [64] which have been trained on a separate dataset [53].

---

[2]https://nda.nih.gov/edit_collection.html?id=19
[3]https://www.oasis-brains.org/

# E Need for regularization

In this section, we qualitatively and quantitatively illustrate that proper regularization of patch-wise *negative-free* multiscale representation learning avoids low-diversity or degenerate decoder representations which hamper downstream segmentation performance. We primarily compare the complete proposed model (*'Ours w/ regularization'*) against an ablation (*'Our ablation w/o regularization'*) that only uses the patch-wise similarity loss and does not use the denoising, variance, covariance, and orthogonality regularization (which corresponds to ablation E from Table 2).

**Figure 14** visualizes activations from layers {6,8,10,13,16} of Table 9 (left) and the sorted singular values of the spatial covariance matrices of multi-layer feature projections from the corresponding layer-wise projector output (right) trained with/without regularization. Given a flattened mid-axial feature projection $F \in \mathbb{R}^{wh \times c}$ (where $c = 2048$ and $wh$ is the vectorized spatial dimensionality), we calculate the spatial covariance as $C = \frac{1}{c} \sum_{i=1}^{c} (z_i - \bar{z})(z_i - \bar{z})^T$, where $z_i \in \mathbb{R}^{wh \times 1}$ is a spatial feature vector and $\bar{z} = \frac{1}{c} \sum_{i=1}^{c} z_i$. The lower the rank of $C$, the lower the spatial variability of representations.

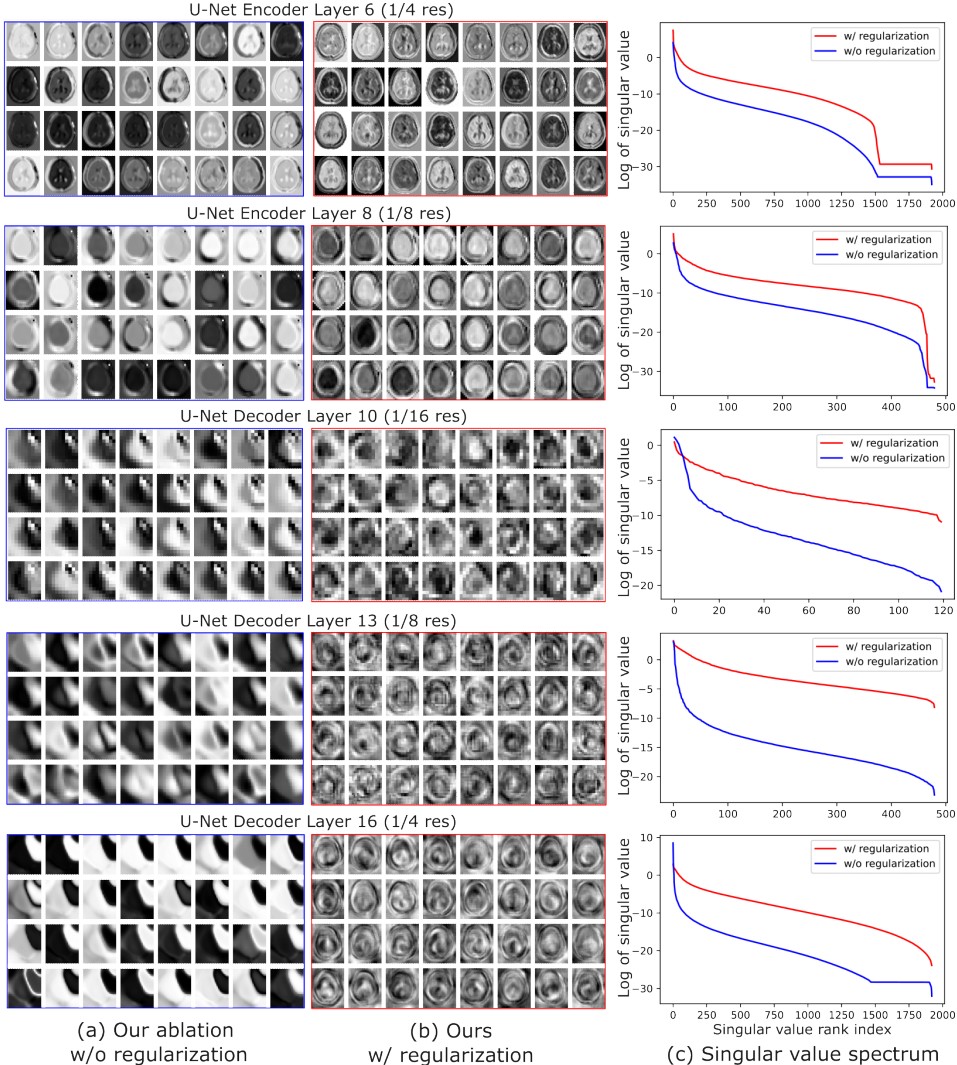

(a) Our ablation w/o regularization

(b) Ours w/ regularization

(c) Singular value spectrum

Figure 14: An illustration of U-Net activations (left) and the log-scale singular values of the spatial covariance matrix $C$ (right), both trained with (red) and without (blue) regularization. While encoder activations are comparable both with and without regularization, the decoder U-Net activations converge to degenerate spatial patterns without regularization (left) and their projections have much lower spatial variability as shown by the singular values of their covariance matrices (right). With regularization, decoder layers avoid degenerate solutions and their spectra indicate higher rank.

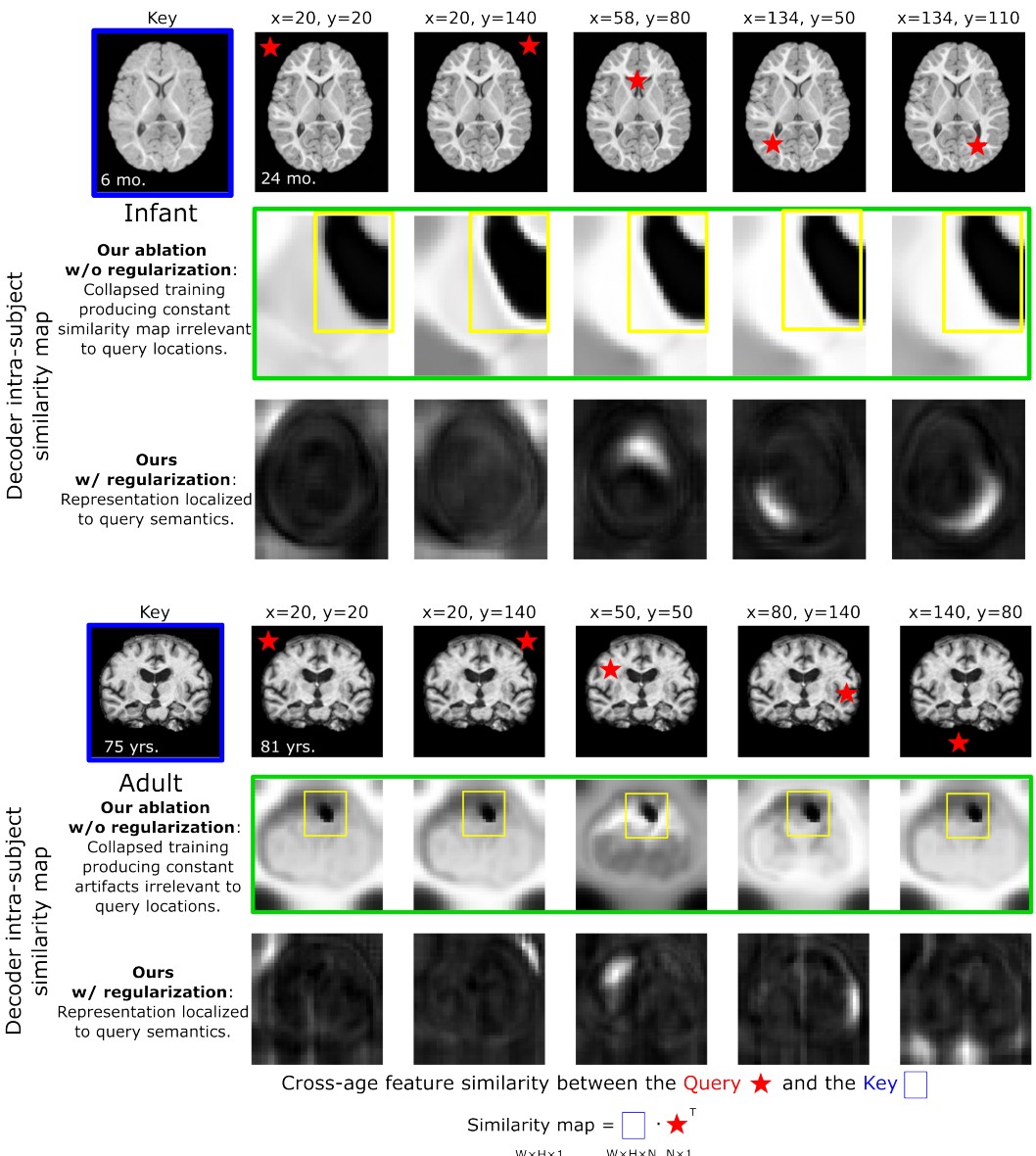

Figure 15: **Varying queries against a constant key.** To complement Figures 1 and 5, this figure qualitatively visualizes the intra-subject similarity maps obtained by querying the projections of decoder layer 16 (from Table 9) from several distinct regions (red stars, columns) and comparing them against the projection of the corresponding decoder activation from a key intra-subject temporal image (blue box, left). Without regularization (rows 2 and 5), the ablation yields intra-subject similarity values with very low spatial diversity across all queries, as indicated by similar appearance across all entries in the green boxes and the spatial artifacts in the yellow boxes. However, given proper regularization (rows 3 and 6), these values converge to semantically-similar regions, which indicate better representations for semantically-driven tasks as indicated by improved downstream segmentation performance.

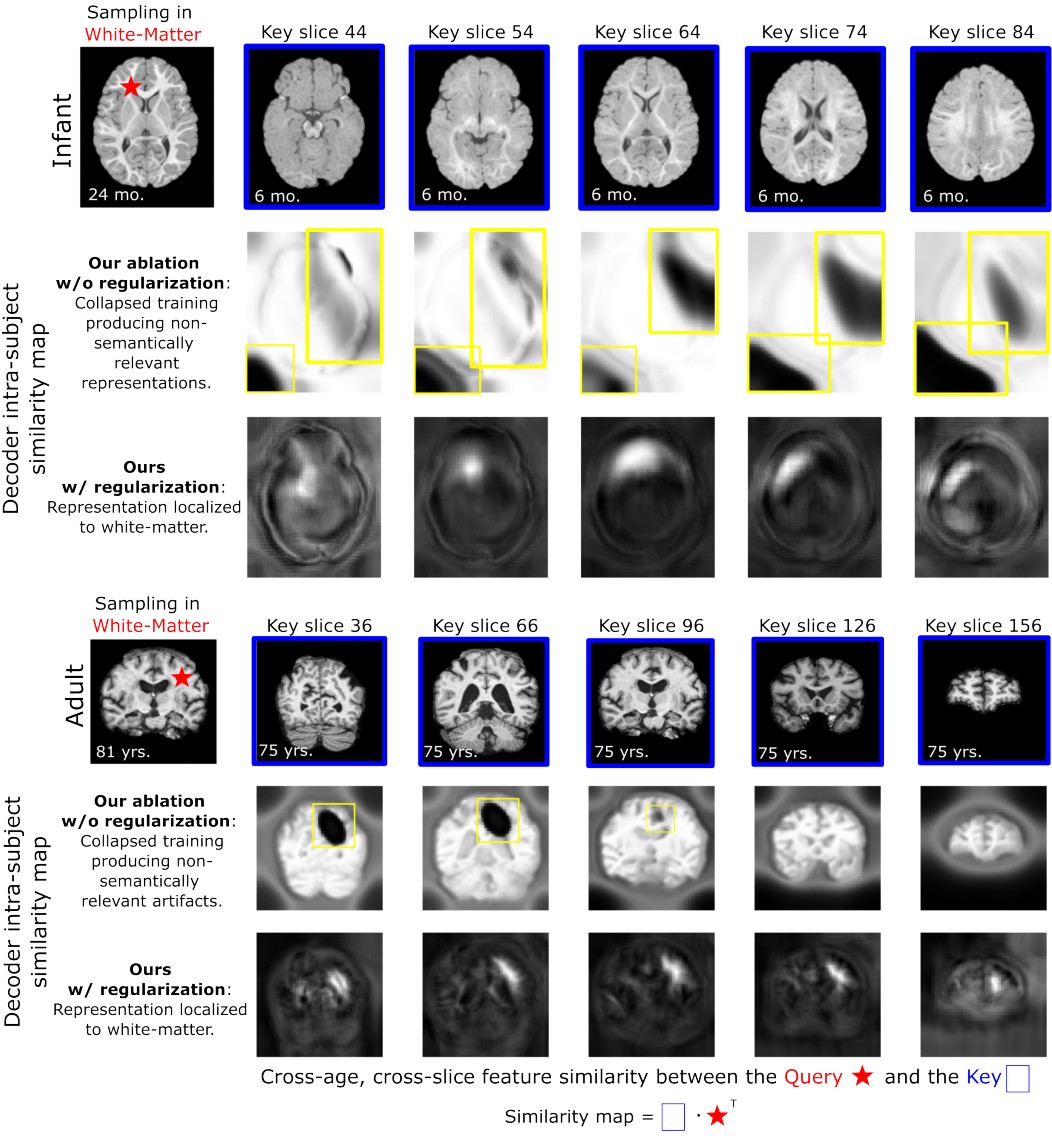

Figure 16: **Varying keys against a constant query.** To complement Figure 15, we now hold constant the query (red star, sampled from white matter) decoder feature projection from layer 19 (of Table 9) and visualize the intra-subject similarity values computed against several distinct key slice projections (blue boxes) from the corresponding decoder representations of an intra-subject temporal image. As in Figure 15, the ablation without regularization (rows 2 and 5) yields degenerate and semantically-unmeaningful similarity patterns (yellow boxes) with low spatial diversity and artifacts across 3D space. With regularization, self-similarity patterns are semantically-coherent in 3D.

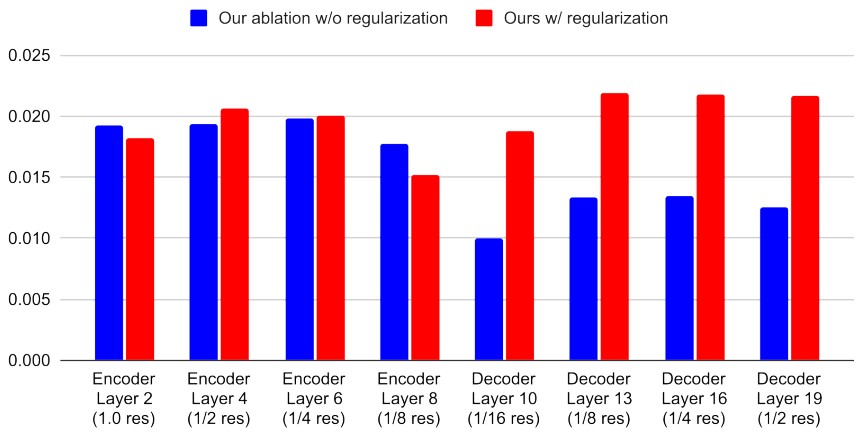

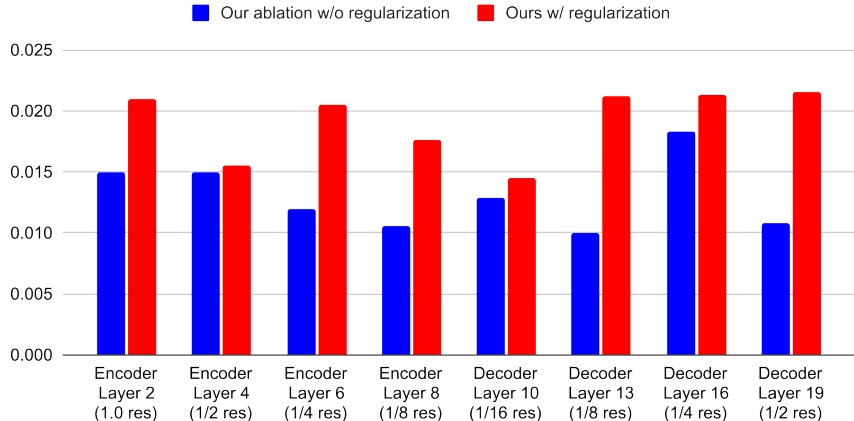

Figure 17: **Variability of local U-Net feature embeddings.** This figure visualizes the standard deviations of channel-wise projection vectors from multiple layers of a U-Net pretrained with (red) or without (blue) regularization. As previously suggested by Figures 14, 15, and 16, decoder projections have significantly lower spatial variability without regularization, indicating low-diversity representations. With regularization, spatial variability is increased which ultimately enables better transfer to downstream tasks such as segmentation.

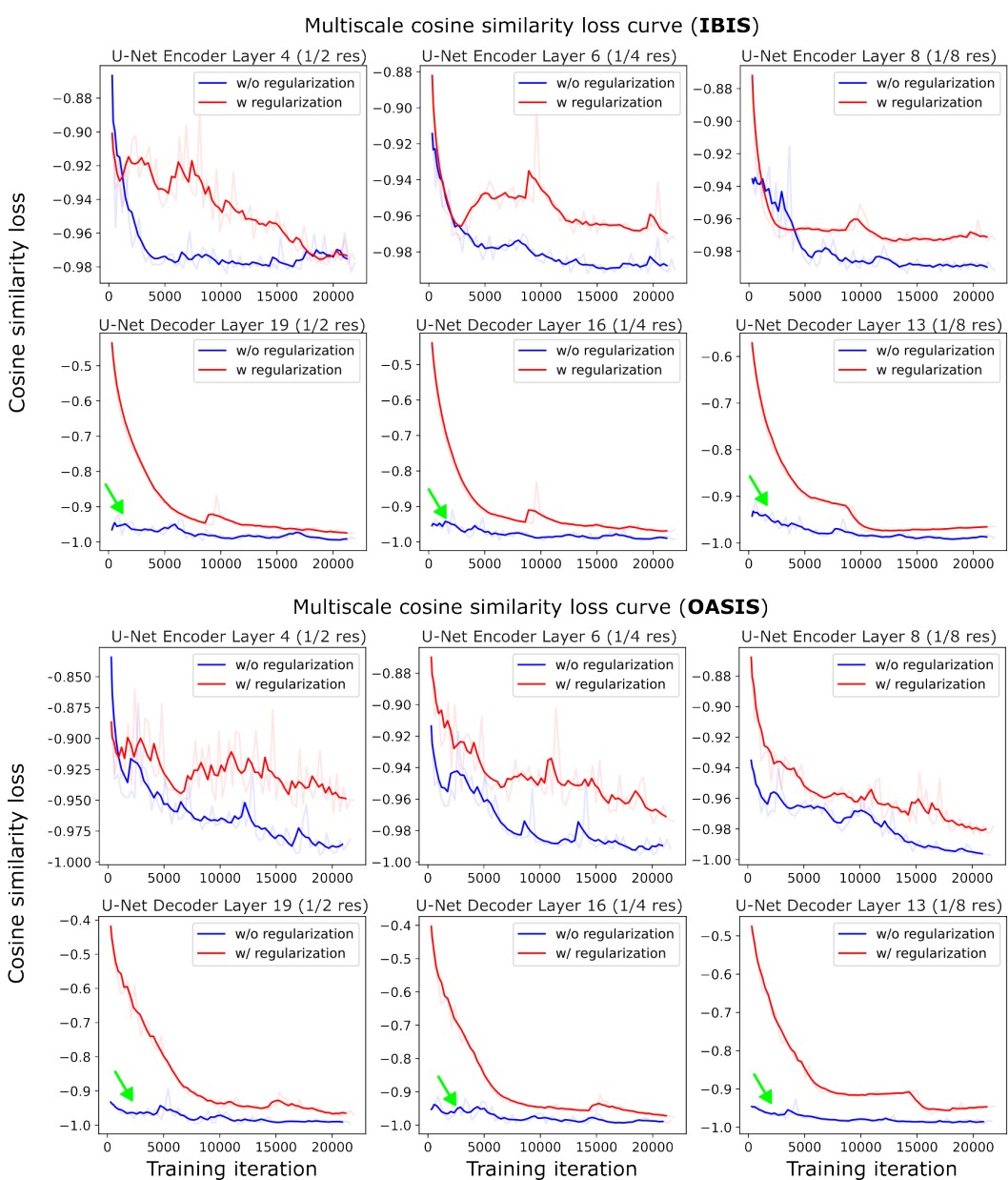

Figure 18: **Multi-layer patch-wise cosine similarity loss** ($\mathcal{L}_{sim}$)**.** Here, we visualize the positive-only patch similarity loss against training steps for a selection of encoder and decoder layers (columns) across both datasets with (red) and without (blue) regularization. During initial training stages, we see that encoder layer losses (rows 1 and 3) gradually decrease in both regularized and unregularized settings. However, decoder layer losses (rows 2 and 4) show near-immediate convergence to a degenerate ideal solution (green arrows) without regularization and these unregularized representations do not transfer well to downstream segmentation as shown by the quantitative results in this paper. This phenomenon is consistent with previous literature on negative-free representation learning (see Figure 2a of [13]).