# OpenReview forum: "Local Spatiotemporal Representation Learning for Longitudinally-consistent Neuroimage Analysis"
_NeurIPS.cc/2022/Conference — NeurIPS 2022 Accept_

### Official Review · Reviewer_de4N · 2022-07-08

**Rating:** 7
**Confidence:** 4
**Soundness:** 3 good
**Presentation:** 3 good
**Contribution:** 3 good

**Summary:**

The paper claims that prior works on self-supervised learning are ill-suited to handle typical longitudinal medical image sequences with problems due to assumptions of i.i.d data and methods that are not able to do per pixel-level predictions. As such the authors propose a number of solutions:

1. a spatiotemporal patchwise similarity loss, which attempts to increase longitudinal similarity between features at similar spatial positions and at all layers of the network
2. enforcing orthogonality of features from the encoder and decoder.
3. encouring variance and decorrelation of decode features
4. denoising reconstruction loss
5. a finetuning term that encourages segmentations to be similar over multiple time points.

The proposed solutions are evaluated on three datasets with longitudinal medical images, in comparison with
a number of previously developed self-supervision methods aswell as standard supervised methods such as UNet
.


**Questions:**

- I have problems appreciating the proposed solutions as I do not understand and not convinced of the problem the authors are trying to address. Specifically the claims of mode collapse, supposedly illustrated in Figure 1. Could the authors do more to describe, quantify and illustrate this problem?

**Limitations:**

- Limitations are reasonably well addressed.

**Strengths And Weaknesses:**

+ Self-supervision in medical imaging and particularly in longitudinal analysis is very relevant and an impo
rtant topic, that in my opinion is not well addressed by existing solutions.

- Clarity, I find the work hard to follow with long sentences that often appear ambigious to me.

- Quality and significance, what is the relevance and significance of the results? Variance over subjects and multiple training results is not presented. What is the relevance outside one-shot-segmentation?

- Soundness, I am not sure the observations and conclusions of mode collapse (Fig.1) is correct. There is little evidence presented. The speculation that this is due to skip connections between the encoder and decoder seems hard to verify from the presented evidence. This speculation is repeated multiple places and used as a design argument in sections "Methods - Orthogonality, Variance and covariance".

---

> ### Author Response · Authors · 2022-08-02
> **Response to Reviewer de4N (part 1/2)**
>
> Thank you for your valuable feedback and for noting the importance of the problem being worked on. We believe that there may have been a few technical and presentation miscommunications on our part and we hope to address them below.
>
> > *Clarity, I find the work hard to follow with long sentences that often appear ambigious to me.*
>
> Thank you for letting us know. We have revised the text for further clarity (with new text marked in green) and would be happy to receive feedback on which specific sections could be made clearer.
>
> > *Variance over subjects and multiple training results is not presented.*
>
> This may be a miscommunication as experimental variances are always reported except in the cases where we additionally report median values.
> - If Table 1 is being referred to, we report *median* scores as a summary statistic across a variety of performance scores and datasets for visual clarity and space purposes. As indicated in the caption, means and standard deviations for the same experiments are reported in the supplementary material (Table 7, reported as `mean(std)`).
> - If the numbers on top of the individual boxes in Figure 4 are being referred to, those are also *median* statistics to add additional information on top of the means and standard deviations which are already encoded in the boxplot y-axis values and box heights. The remaining quantitative results all report means and standard deviations, except where noted to be the median.
>
> If we missed any other quantitative summary, please let us know and we would be happy to add it.
>
> > *What is the relevance and significance of the results?*
>
> We respectfully posit that developing a self-supervised spatiotemporal representation learning framework which generalizes to every setting of annotation-availability (one/few/full-shot segmentation, see below) is a significant and relevant result for the biostatistics of neuroanatomical disorders. This is demonstrated by both:
> - Improved downstream segmentation performance (a core medical imaging task) and;
> - Improved longitudinal consistency in downstream segmentations (crucial to biostatistics).
>
> Further, these improvements are shown to generalize across diverse neuroimaging datasets showing heterogeneous patterns of *both* degeneration (elderly adults with/without Alzheimer's Disease) and development (infants with/without Autism), which has not been previously demonstrated to our knowledge. Lastly, we note that our representation learning method is general-purpose for voxel-level image time-series tasks and can potentially be used for other applications in future work as well.
>
> > *What is the relevance outside one-shot-segmentation?*
>
> Alongside the one-shot segmentation improvements provided in the main text, we also demonstrated improvements over every baseline in the **few-shot** (10% labeled data) and **fully-supervised** (100%) segmentation settings in Section A of the Suppl. Materials in Tables 3 and 4. This could have been communicated better on our part and we now accordingly emphasize these additional suppl. experiments in the main paper.
>
> ## Major concern & response:
> > *I have problems appreciating the proposed solutions as I do not understand and not convinced of the problem the authors are trying to address. Specifically the claims of mode collapse, supposedly illustrated in Fig1. Could the authors do more to describe, quantify and illustrate this problem?*
>
> Thank you for this question. We will try to recontextualize the problem and our solutions in our response here alongside providing several new illustrations and quantifications in **Section E** of the Supplementary Material.
>
> **What problem are we trying to solve?** Our main objective is to leverage longitudinal scans to learn useful *multi-scale* representations in all layers of an image-to-image architecture in a self-supervised manner. If properly trained, these representations should generalize to downstream tasks such as segmentation and provide better longitudinal consistency. Importantly, no current method attempts simultaneous *multi-scale non-i.i.d. spatiotemporal* representation learning  to our knowledge.
>
> **Where is the miscommunication?** Crucially, the *mode collapse* problem in the initial submission occurs in an **ablation** of our method. This ablation demonstrates that naive extension of negative-free representation learning methods (needed to avoid false negatives with unsupervised contrastive pretraining) to multi-layer patchwise training may lead to degenerate solutions. We follow the representation learning literature [Jing22] and define (in our context) collapsed/degenerate solutions to be the setting where internal patch representations show low-diversity and no semantic coherence. This phenomenon is well established in the negative-free global (i.e., one embedding per image) representation learning literature and always needs regularization (e.g., [Chen21], BYOL, WMSE, VICReg, BarlowTwins, etc).
>
> (cont’d)

---

> > ### Author Response · Authors · 2022-08-02
> > **Response to Reviewer de4N (part 2/2)**
> >
> > (cont’d)
> >
> > **What are our solutions doing?** To rectify the problems of this ablation, we introduce several forms of regularization designed to increase the spatial and inter-channel variability of representations and ultimately yield diverse representations for neuroanatomical structures. To explain **Figure 1**, in row B (yellow box), the intra-subject similarity maps are either constant or have no semantic relevance. On adding proper regularization to this ablation in row C (green box), the decoder self-similarity maps localize to frontal lobe white matter as expected. Similar trends are observed in Supplemental Figure 5.
> >
> > For clarity, we now replace the term “mode collapse” with *low-diversity embeddings* (with artifacts, when relevant).
> >
> > **Can we do more to describe, quantify, and illustrate this problem?** Yes, thank you for this suggestion. We now provide further qualitative and quantitative evidence of low diversity embeddings and artifacts in that ablation in **Supplementary Material Section E**:
> > - **Figure 14** visualizes U-Net activations without/with regularization alongside the singular values of the sample covariance matrices (inspired by [Jing22]) of the spatial feature projections.
> >     - In the singular value plots, the representation space is shown to collapse/have a lower rank without regularization, as indicated by the vanishing singular values. This rapid spectral decay is lessened by the proposed regularization indicating representations of higher spatial variability.
> >     - While the encoder activations (rows 1, 2; cols a, b) qualitatively resemble typical deep network activations with or without regularization, the decoder activations (rows 3–5; cols a, b) converge to degenerate spatial patterns with artifacts without regularization.
> > - In **Figure 15**, we query a diverse range of neuroanatomy and background locations. In the yellow and green boxes for the unregularized ablation, the self-similarity value does not appreciably change or have anatomical relevance regardless of the query. These problems are alleviated with the proposed method.
> > - In **Figure 16**, we fix the query and compute their similarity to various 2D intra-subject key slices in 3D space. We find that the artifacts and low-diversity embeddings are consistent across 3D space in the unregularized ablation (yellow boxes).
> > - In **Figure 17**, we plot the standard deviation of the spatial projection vectors and show that the unregularized ablation has low variance in the decoder layers. With regularization, the decoder layers produce higher spatial variability.
> > - In **Figure 18**, we plot the cosine similarity loss values with/without regularization. As indicated by the green arrows, the decoder similarity losses without regularization converge to degenerate solutions and achieve a trivially-true ideal loss as is common with negative-free approaches (please see Figure 2a of [Chen21]). With regularization, both encoder and decoder losses converge more gradually, which typically empirically corresponds to more useful representations downstream.
> >
> >
> > > *The speculation that this is due to skip connections between the encoder and decoder seems hard to verify from the presented evidence. This speculation is repeated multiple places and used as a design argument in sections "Methods - Orthogonality, Variance and covariance".*
> >
> > As described above, we demonstrate that while the losses, feature maps, standard deviations, and singular values of the encoder layers typically achieve comparable results with or without regularization, the *decoder* activations and projections all typically converge to degenerate solutions without regularization early in training. Our speculation (based on the empirical trends presented in Figures 14-18) is that skip connections enable the decoder loss to be trivially minimized using good representations from the encoder (Figure 18), and that this trivial minimization does not enforce learning semantically-relevant decoder representations.
> >
> > However, we agree that this speculation should have been better supported in the initial submission and we hope that this is addressed by the new empirical support presented in **Supplemental Sec. E**. We have also revised the presentation of the methods to account for this and have clarified the speculative aspects. Please let us know if any detail remains unclear
> >
> > Lastly, we note that the ultimate success of representation learning methods is most commonly empirically measured by higher performance on downstream tasks. In our context, decoder regularization is demonstrated to learn representations which transfer better across multiple segmentation tasks with several measures such as performance and consistency over unregularized ablations and multiple existing baselines.
> >
> > ## References
> > [Jing22] Jing, et al. "Understanding Dimensional Collapse in Contrastive Self-supervised Learning." ICLR22
> >
> > [Chen21] Chen & He. "Exploring simple siamese representation learning." CVPR21

---

### Official Review · Reviewer_r9Zh · 2022-07-08

**Rating:** 7
**Confidence:** 4
**Soundness:** 3 good
**Presentation:** 3 good
**Contribution:** 3 good

**Summary:**

This paper tackles self-supervised learning for longitudinal medical images by designing loss functions that consider the temporal evolutions of scans taken from the same subject with applications in downstream segmentation. A U-net architecture is pre-trained via a combined loss function that (i) maximizes the similarity between encoder features from corresponding local patches acquired at different times from the same subject, (ii) minimizes the similarity between encoder and decoder features at the same resolution, as well as at different channels to combat mode collapse and improve generalization, (iii) minimizes the reconstruction error between different augmentations of the same scans. Following pre-training, one-shot segmentation fine-tuning is augmented again with temporal consistency by maximizing the Dice score between segmentation predictions at different times from the same subject. Resulting method outperforms both randomly initialized U-Net, as well as several state-of-the-art pre-training methods with and without benefiting temporal information.

**Questions:**

Following works also tackle self-supervised learning in longitudinal brain imaging. Can the authors address their differences with the current work?

Wei et al. “Consistent Segmentation of Longitudinal Brain MR Images with Spatio-Temporal Constrained Networks”, MICCAI, 2021

To et al. “Self-supervised Lesion Change Detection and Localisation in Longitudinal Multiple Sclerosis Brain Imaging”, MICCAI, 2021


**Limitations:**

Limitations and potential impacts have been addressed.

**Strengths And Weaknesses:**

Strengths: Self-supervised learning from longitudinal medical imaging is an important problem due to costly annotations. The paper is well-written, particularly in experiment setup. Generalization of state-of-the-art autoencoder pre-training principles to temporal consistency is novel.

Weaknesses: Motivation part of the introduction could be expanded and improved with some citations.

---

> ### Author Response · Authors · 2022-08-02
> **Response to Reviewer r9Zh**
>
> Thank you for the encouraging evaluation and for the missing references!
>
> > *Weaknesses: Motivation part of the introduction could be expanded and improved with some citations. Following works also tackle self-supervised learning in longitudinal brain imaging.*
>
> We agree, thank you for letting us know of these relevant papers - we have now added them to the submission. We further discuss differences between our submission and their work in detail below:
>
> > *Can the authors address their differences with the current work? To et al. “Self-supervised Lesion Change Detection and Localisation in Longitudinal Multiple Sclerosis Brain Imaging”, MICCAI, 2021*
>
> To, et al.’s work is a specialized framework for synthetic image and pseudo label generation for *change detection* of multiple sclerosis lesions (with homogeneous intensity appearance) in MRI time-series. They do so by training a VAE on a sequence of images with no changes and then augment the VAE reconstructions to simulate changing lesions. They then use these augmented images to train a change detection segmentation network. As such, their goals and tasks are distinct from ours as we do not work on change detection of MS lesions or on synthetic data generation.
>
> Instead, we focus on general-purpose unsupervised spatiotemporal representation learning, with downstream generalization demonstrated to one-shot segmentation of multiple brain structures (and not just lesions) in the main paper and few-shot and fully supervised segmentation extensions in the supplementary material.
>
> > *Can the authors address their differences with the current work? Wei et al. “Consistent Segmentation of Longitudinal Brain MR Images with Spatio-Temporal Constrained Networks”, MICCAI, 2021*
>
> Wei, et al. develops a two-stage segmentation-focused framework that does not work on self-supervised representation learning. To summarize their method:
> - In stage 1, they first train a fully-supervised segmentation network on a fully-annotated cross-sectional MRI dataset with hundreds of subjects.
> - In stage 2, they finetune this network on unlabeled longitudinal data using specialized losses to make its segmentation predictions more temporally consistent. These losses include consistency losses over features, volume counts, labels, and penalizing deviations from a quadratic volume count model (pre-fitted on a large-scale supervised dataset).
>
> To detail the distinctions between our submission and their framework:
> - Their method requires access to a large annotated cross-sectional dataset for fully-supervised segmentation pretraining. We instead develop general-purpose self-supervised representation learning methods on unlabeled longitudinal images and then finetune on downstream tasks such as one-shot, few-shot, and fully-supervised segmentation.
> - Some of their stage-two finetuning losses assume access to information which may not broadly generalize across various types of longitudinal studies. For example, they assume access to 3+ repeat images per subject, access to a fully-annotated large dataset for fitting a quadratic volume model, and constant image intensity across time via their MSE-based feature consistency loss (which is inapplicable to infant studies such as ours where brain intensity strongly changes with time, for example).
> - Their segmentation consistency loss is similar to ours at a very high-level, with the primary differences being that we use segmentation consistency as a *regularizer* alongside a main segmentation loss while they minimize segmentation consistency as a primary loss. Minimizing a segmentation consistency loss alone as in their work may potentially have a degenerate solution where the network predicts the same structure regardless of age. Their work appears to avoid this degenerate solution by using a quadratic volume model regularizer - however, this regularizer requires a large-scale labeled database for tuning the per-label quadratic model which is inapplicable in our setting.
>
> To reiterate, these references have been added and we thank you for your time and feedback. We would be happy to address any further questions and/or comments.

---

### Official Review · Reviewer_xiZS · 2022-07-11

**Rating:** 7
**Confidence:** 2
**Soundness:** 3 good
**Presentation:** 3 good
**Contribution:** 3 good

**Summary:**

The paper proposes a self-supervised deep-learning framework for image-to-image translation tasks, such as segmentation, that accommodates and fully exploits longitudinal data.  Specifically the method provides a mechanism to impose consistency in output across multiple points from the same individual and simple regularisation terms to avoid common problems with other methods, such as mode collapse. The authors compare the method against baselines in two distinct neuroimaging segmentation tasks, which nicely demonstrate the additional power afforded by imposing longitudinal consistency.

**Questions:**

Broadly happy with it apart from the minor issues raised above.

**Limitations:**

Seems fine.

**Strengths And Weaknesses:**

Strengths

- The work addresses a persistent challenge and shows a significant advance.

- The experimental work is well thought out and showcases nicely the potential of the idea.

Weaknesses

- (minor) In places the writing could be made more accessible to the non-specialist and organised a bit better. Highlighting specifically what the reader is looking for in Figure 1 would help interpret that figure.  The task isn't specified in the OASIS paragraph (starting line 202). These things can be figured out but the authors could make the reader's job easier.

---

> ### Author Response · Authors · 2022-08-02
> **Response to Reviewer xiZS**
>
> Thank you for the positive feedback!
>
> > *In places the writing could be made more accessible to the non-specialist and organised a bit better. Highlighting specifically what the reader is looking for in Figure 1 would help interpret that figure. The task isn't specified in the OASIS paragraph (starting line 202). These things can be figured out but the authors could make the reader's job easier.*
>
> We agree. We have now added arrows and boxes for emphasis to Figure 1 and revised its caption to make the figure easier to interpret. The task has been described in the referenced OASIS paragraph and we have also revised the overall paper for clarity where possible.
>
> We would be happy to hear of any other aspects which could be further improved and thank you again for your time.

---

### Author Response · Authors · 2022-08-02
**Overall Response**

We thank the reviewers for their time and expert feedback which has been used to improve the revised submission.

As a summary of the reviews, we were happy to see that the submission was found to tackle an important problem [`xiZS, r9Zh, de4N`] with well formulated experiments [`xiZS, r9Zh`] which show a significant advance [`xiZS`] and novel generalization [`r9Zh`]. Concerns raised included non-specialist accessibility [`xiZS`], adding more related work [`r9Zh`], and questions w.r.t. the claims, presentation, and relevance of the results [`de4N`].

To these ends, we address these concerns in the revision (with new text marked in green) and individual responses below. Major changes include:
- Based on Reviewer `de4N`’s concerns, we have added a new section (**Section E/Need for regularization**) to the Supplementary Material which contains several new visualizations and quantifications of the *mode collapse* problem with one of our ablations which is addressed by the proposed regularization.
- Further, several technical and experimental clarifications are made in the individual response to Reviewer `de4N` and the paper has been updated to reflect these clarifications where applicable.
- The presentation has been made more accessible [`xiZS`] and ambiguous sentences have been made clearer [`de4N`].
- The citations suggested by Reviewer `r9Zh` are now discussed in the submission.
- We have added the appendices to the main paper PDF for easier reference.

Again, we deeply appreciate the feedback and are happy to receive any further questions, comments, and/or suggestions for improvements.

---

### Meta-Review · Area_Chair_gtDL · 2022-08-26

**Recommendation:** Accept
**Confidence:** Certain

**Metareview:**

The paper proposes a self-supervised deep-learning framework for image-to-image translation tasks, such as segmentation, that accommodates and fully exploits longitudinal data. Specifically the method provides a mechanism to impose consistency in output across multiple points from the same individual and simple regularisation terms to avoid some problems common with other methods, such as mode collapse. The authors compare the method against baselines in two distinct neuroimaging segmentation tasks, which nicely demonstrate the additional power afforded by imposing longitudinal consistency.

The reviews overall reported that the submission tackles an important problem, presents well formulated experiments, and shows a significant advance over the SOTA.

The reviews raised concerns raised included non-specialist accessibility, adding more related work, and questions w.r.t. the claims, presentation, and relevance of the results, and particularly about the mode collapse problem. The authors have addressed these concerns, in particular by adding a new section (Section E/Need for regularization) to the Supplementary Material which contains several new visualizations and quantifications of the mode collapse problem (from ablation experiments).
The paper has been updated to reflect some  clarifications required by reviewer de4n. Some citations suggested by Reviewer r9Zh are now discussed in the submission.

As it is, the paper meets all conditions for acceptance at NeurIPS 2022.


**Award:**

Yes

---

### Decision · Program_Chairs · 2022-09-14

Accept